# Causal Explanations for Human Understanding in Deep Neural Policies

## Abstract

Explainable deep learning models are important for the development, certification, and adoption of autonomous systems. Yet, without methods to quantify causal relationships between explanations and actions, interpretability remains correlational. Furthermore, explanations typically address lower-level actions. This poorly serves human understanding, which benefits from higher-level abstractions, and underactuated robotics, whose behaviors often require richer descriptions. To address these gaps, we introduce Causal Concept Wrapper Network (CCW-Net), a general representation-learning method across differentiable architectures that adapts mediation analysis from fields such as economics, medicine, and epidemiology to align the causal effects of abstract, information-rich explanations with policy actions. CCW-Net expands the expressiveness of prior work in both explainable deep learning and mediation analysis allowing each explanation to serve as a mediator encoding both its presence and context-based expression. In a high-fidelity, underactuated aircraft formation task, CCW-Net produces high-level explanations that are both interpretable and quantifiably causal without degrading task performance. We demonstrate CCW-Net across diverse architectures including capsule networks with dynamic routing, modified concept bottleneck models, and cross-attention mechanisms. Notably, we present the first adaptation of capsule networks to sequential decision-making in robotics. This breadth shows that CCW-Net applies broadly across neural network architectures, offering a general path toward transparent and trustworthy autonomy.

## 1 Introduction

Learned policies in autonomous systems are moving from research labs into high-stakes, real-world settings such as aviation (Pope et al., 2022; Ward, 2023), driving (Phan-Minh et al., 2023), and robotics (Tang et al., 2025). In task and safety-critical domains such as these, interpretability is becoming a necessary component (Rudin, 2019; Atakishiyev et al., 2025). Developers need it to troubleshoot (Kenny et al., 2024), testers need it to verify (Rountree et al., 2021; Mahmud et al., 2024a), and users need it to understand (Sanneman & Shah, 2022). Yet many neural policies remain black boxes: they are not interpretable in a way that is useful for people in the task at hand.

Recent concept-based interpretability offers promise by structuring decisions through human-interpretable concepts (Koh et al., 2020; Echterhoff et al., 2024). However, prior work typically represents concepts as scalars (Madumal et al., 2020; Koh et al., 2020; Kenny et al., 2024), for example, a "left turn" in driving or the "build supply depot" action in Starcraft II. Scalars are useful for indicating the presence or strength of a concept, but they cannot capture the higher-dimensional structure needed to communicate complex maneuvers or concepts that can manifest themselves in many ways. Additionally, increased abstraction allows people to convey information in denser, richer units that balance user workload and understanding (Sanneman et al., 2024). In this work, we extend concept representation from scalars to vectors, enabling each concept to encode both *whether* it exists and *how* it is expressed bringing concept-based explanations closer to the richness of human reasoning (Tucker et al., 2022).

Moreover, the relationship between concepts and actions remains largely correlational, i.e., a policy might activate a concept without another concept pathway causing the observed action, leading to spurious explanations (Zhou et al., 2022). A vehicle policy might activate a "passenger pickup" con-

cept but it does not mean that no other concept caused the observed action. The model may appear to explain its actions when in reality relying on spurious correlations and confounding patterns in the data. To be useful in troubleshooting, certifying, and understanding explanations must capture causal effects or what would happen if we intervened to change a concept while holding others fixed (Pearl, 2012). Correlation, no matter how sophisticated, does not address causal questions (Pearl, 2009). Misleading explanations can obscure risks, prevent effective troubleshooting, and erode user trust (Wang et al., 2022). For safety and task-critical autonomy we must transition from correlation to causation.

We address these challenges with Causal Concept Wrapper Network (CCW-Net), a training method that adapts causal mediation analysis, well established in fields such as economics and epidemiology (Celli, 2022; Lee et al., 2021), to the design of interpretable policies. CCW-Net estimates the causal effect of each concept on actions and trains a policy to align its concept representations with these effects. In doing so, it produces explanations that are both human-meaningful and quantifiably causal while expanding the representational capacity of concepts beyond scalars to high-dimensions. Viewed another way, CCW-Net can be interpreted as a form of causally inspired representation learning: the concept encoder is trained not only to predict human-defined concepts but also to organize its latent representation so that each concept vector accounts for its empirically estimated causal effect on the action, making concepts locally necessary and sufficient for components of the policy's behavior.

**Main contributions**   Our contributions are fourfold:

1. **Causally Guided Representation Learning:** A training objective aligning concept-to-action sensitivity with causal estimates, yielding explanations that are locally necessary and sufficient.

2. **Causal Alignment through Mediation Analysis:** Adoption of mediation analysis to imitation learning to estimate causal targets from logged trajectories and align concept-action attributions.

3. **Concepts as Vectors:** Extend concept bottleneck models to represent concepts as vectors capturing contextual structure enabling abstract, human-aligned explanations.

4. **Architecture Agnostic:** A general framework broadly applicable to any differentiable policy head, independent of architecture choice and task domain, enabling broad integration.

## 2   RELATED WORK

We now briefly discuss a few related research directions. Additional discussion is in App. B.

**Concept-based explanations**   The idea of using concepts to generate explanations of AI systems is widely explored (Kim et al., 2018; Alvarez-Melis & Jaakkola, 2018; Koh et al., 2020; Bai et al., 2023; Achtibat et al., 2023; Tan et al., 2024). These methods have found applications in variety of fields, including biomedical applications (Graziani et al., 2018; Clough et al., 2019; Yeche et al., 2019), scientific research (Sprague et al., 2019; Yang et al., 2024), game-playing systems (Lovering et al., 2022; Tomlin et al., 2022; Schut et al., 2025), planning agents (Kazhdan et al., 2021; Qian et al., 2024), etc. These concepts can come from various sources: as input from human experts (Ghandeharioun et al., 2022), or by extracting them from labeled data (Ghorbani et al., 2019; Yeh et al., 2020).

**Concept-based interpretable policies**   Concept Bottleneck Models (CBMs) (Koh et al., 2020) and their adaptations to control (Kenny et al., 2023; 2024) structure decisions through interpretable concepts. However, these methods typically rely on scalar representation, which capture concept presence but lack the expressiveness to encode context-dependent execution (e.g. the *manner* of a maneuver), and do not enforce causal relationships between concepts and actions. CCW-Net addresses this via causally guided representation learning. In contrast to prior art, we shape the concept encoder's latent geometry using interventional effect estimates rather than predictive correlations. This encourages directions in the concept space to become locally necessary and sufficient for components of the policy's action.

**Causal mediation analysis** In parallel, causal inference methods have developed sophisticated tools for establishing causal relationships from observational data. Other fields such as epidemiology, economics, and medicine have developed mature tools for causal mediation analysis (MacKinnon, 2008; Imai et al., 2010; VanderWeele, 2015). Causal mediation analysis decomposes treatment effects into pathways through mediators (Pearl, 2009; Vansteelandt & Daniel, 2017; Loh et al., 2022), with the interventional indirect effect (IIE) quantifying how much of a treatment's effect operates through specific mediators. These methods are identifiable from observational data under standard ignorability conditions but have not been adapted to establish causal relationships between learned concept representations and model outputs in deep learning. In this direction, our contribution is to adapt these causal tools to imitation learning and extend their treatment of individual mediators from scalars to vectors.

**Causal attribution methods in deep learning** Finally, a wide range of attribution methods aim to explain deep networks by linking inputs or intermediate features to outputs. Examples include saliency maps (Simonyan & Zisserman, 2015; Selvaraju et al., 2017), perturbation methods (Ribeiro et al., 2016; Lundberg & Lee, 2017), and gradient based techniques (Smilkov et al., 2017; Sundararajan et al., 2017). While these methods can highlight what features are associated with a decision, they generally remain correlational. For instance, ablating a feature may change a prediction, but this does not establish that the feature causally drives the outcome (Adebayo et al., 2018; Zhou et al., 2022). In safety and task-critical autonomy, such correlational explanations can be misleading. Our approach differs by explicitly grounding attribution in estimated causal effects. We not only compute how actions respond to concept changes, but also align these sensitivities with causal effects estimated from data.

## 3  PROBLEM FORMULATION

In this work, we address the challenge of training interpretable neural policies that provide causally grounded explanations for their actions. Given a set of trajectories and expert human-defined concepts, we seek to learn a policy that: (1) matches expert performance, (2) reasons over interpretable concepts, and (3) ensures that concept–action relationships reflect true causal effects rather than spurious correlations.

**Input:** We assume access to: (1) a pretrained black-box policy $f$ that achieves good task performance but lacks interpretability, and (2) expert trajectories $\mathcal{T} = \{(X_i, Y_i)\}_{i=1}^N$ where $X_i \in \mathcal{X}$ are observation states (e.g., sensor readings, game states) and $Y_i \in \mathcal{Y}$ are expert actions, and (3) concept labels $c^{(\ell)}$ indicating which human-interpretable concepts are active for each sample.

**Desired output:** An interpretable policy $\pi_\omega : \mathcal{X} \to \mathcal{Y}$ that achieves expert-level task performance, provides concept-based explanations for its predicted actions $\hat{Y}$, and ensures these explanations reflect causal relationships between concepts and actions. We jointly optimize imitation, concept classification, and causal alignment (Sec. 5.3.3).

**Assumptions:** We assume that (i) the provided human concepts capture the key decision-making factors for the task; (ii) the expert demonstrations provide reliable ground truth for both task performance and concept labeling; and (iii) the standard ignorability conditions hold (Sec. 5.3.1 and App. C).

## 4  PRELIMINARIES

**Structural causal models** A structural causal model (SCM) (Pearl, 2009; Peters et al., 2017) consists of a set of equations, $X_i = k_i(pa_i, u_i), i = 1, \ldots, m$, where each equation represents an autonomous mechanism that determines the value of exactly one distinct variable; $X_i$ and $u_i$ are the $i$-th random variable and its corresponding error term, respectively. The function $k_i$ represents the causal mechanism generating $X_i$, and $pa_i$ denotes the set of variables that directly cause $X_i$, i.e., are parent variables of $X_i$.

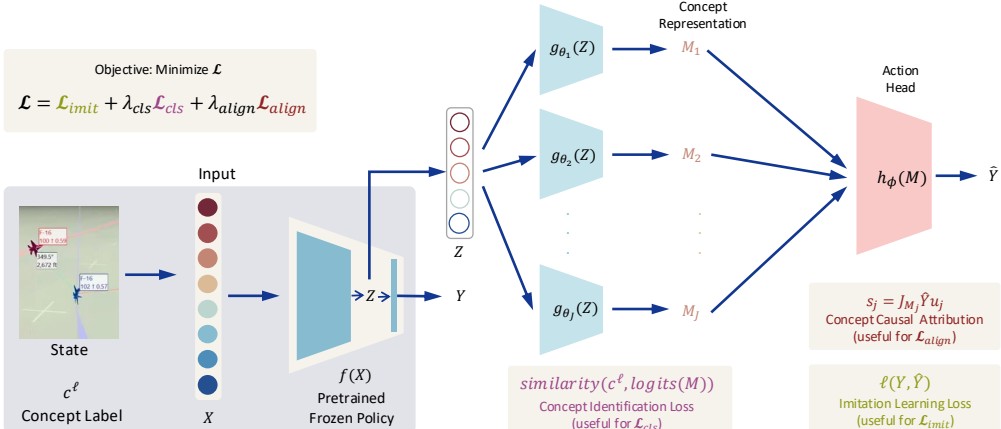

Figure 1: **CCW-Net Architecture.** CCW-Net wraps a frozen backbone $f$ with a concept encoder $g_\theta$ that transforms latents $Z$ into human-interpretable vector concepts $M$, and a differentiable head $h_\phi$ that maps concepts to wrapper actions $\hat{Y}$. CCW-Net estimates per-concept causal targets $IE_j$ from expert trajectories, computes local concept-to-action sensitivities $s_j = J_{M_j} \hat{Y} u_j$, and aligns them with a masked cosine loss. Vector concepts encode both the strengths and context-dependent expression of each concept, enabling faithful, human-interpretable explanations while preserving task performance.

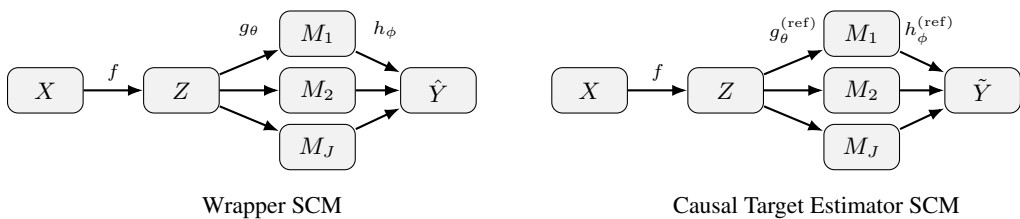

Figure 2: Two SCMs used in CCW-Net. Left: Wrapper SCM used at deployment. The model routes through concepts only, so there is no direct $Z \to \hat{Y}$ path by construction. This SCM represents CCW-Net's architecture in Fig. 1. Right: Causal target estimator with frozen reference components. Interventional causal targets are computed by swapping one concept vector $M_j$ with a matched reference mediator (serving as counterfactual concept values) based on $X$ and concept labels.

**Mediation analysis** Mediation analysis (Pearl, 2012) decomposes the causal effect of a treatment $A$ on an outcome $Y$ into pathways that operate through intermediate variables called mediators $M$. For a treatment $A$, outcome $Y$, and mediators $M = (M_1, \ldots, M_J)$, the total effect *TE* decomposes as $TE = DE + IE$, where *DE* are the direct effects, and *IE* are the indirect effects. For multiple concepts, we can further decompose indirect effects as $IE = \sum_j IE_j$ for all concepts $j$. Furthermore, following Loh et al. (2022), we relax assumptions on the causal ordering of mediators in the causal graph by introducing an interaction term, $IE_\mu$ (Loh et al., 2022). Therefore the causal decomposition of total effects is $TE = \sum_j IE_j + IE_\mu$.

**Imitation learning** Imitation learning trains a parametric policy $\pi_\omega$ to replicate expert behavior from demonstrated state-action pairs $\{(X_i, Y_i)\}$ from the trajectories $\mathcal{T}$ (Zare et al., 2024). The policy parameters $\omega$ are optimized by minimizing the expected imitation loss $\min_\omega \frac{1}{N} \sum_{i=1}^N \ell(\pi_\omega(X_i), Y_i)$, where $\ell(\cdot, \cdot)$ measures the discrepancy between predicted and expert actions. This supervised learning approach assumes the expert demonstrations are optimal or near-optimal for the task, enabling the learned policy to reproduce expert performance without requiring explicit reward engineering.

## 5 CAUSAL CONCEPT WRAPPER NETWORK (CCW-NET)

To solve the problem described in Sec. 3 we introduce Causal Concept Wrapper Network (CCW-Net), a causally inspired representation learning method that enhances a pretrained black-box deep neural policy with human-interpretable explanations that are quantifiably causal to the policy's actions. CCW-Net wraps a frozen backbone policy with a concept module and a new policy head, and is trained to (i) imitate the expert, (ii) predict human-interpretable concept labels, and (iii) align each concept's local effect on the action with a causal target estimated from logged expert trajectories.

Concept wrapper architectures enhance pretrained policies with human-interpretable explanations by inserting an intermediate concept representation layer (Kenny et al., 2023; 2024). We leverage this idea to develop CCW-Net as shown in Fig.1. Given a frozen policy network $f : \mathcal{X} \to \mathcal{Y}$ mapping observations $X$ to actions $Y$, we extract the latent vector $Z = f(X)$ and feed it to a concept encoder $g_\theta$ that outputs vector concepts $M = (M_1, \ldots, M_J)$. A new action head $h_\phi$ maps concepts $M$ to wrapper actions $\hat{Y} = h_\phi(M)$, while the backbone $f$ remains frozen.

### 5.1 STRUCTURAL CAUSAL MODELS AND NOTATION

CCW-Net uses two structural causal models (SCMs), shown in Fig. 2. The first SCM, the *wrapper SCM*, represents how the deployed model maps observations to actions through learned concept vectors. The second SCM, the *causal target estimator SCM*, is used only during training to estimate causal effect targets from logged trajectories using frozen snapshots of the concept encoder and policy head. The two SCMs have the same structure but differ in which components are fixed. We require the target estimator SCM to provide a stable, frozen causal reference frame $(g^{(\mathrm{ref})})$ against which the learning policy can align its gradients.

**Wrapper SCM** At inference the CCW-Net wrapper computes

$$Z = f(X), \qquad M = g_\theta(Z), \qquad \hat{Y} = h_\phi(M).$$

This SCM contains only the mediated path $X \to Z \to M \to \hat{Y}$, with no $Z \to \hat{Y}$ edge, so the direct effect is *zero by construction*. During training we compute a local effect for each concept vector via the directional Jacobian

$$s_j = J_{M_j} \hat{Y} \, u_j, \qquad u_j = \frac{M_j}{\|M_j\| + \varepsilon},$$

and train CCW-Net so that these effects match the causal targets estimated in the second SCM. The wrapper SCM reflects the architecture in Fig. 1.

**Causal target estimator SCM** To estimate causal effects from logged trajectories, we use frozen, cross-fitted snapshots of the concept encoder and policy head:

$$Z = f(X), \qquad M = g_\theta^{(\mathrm{ref})}(Z), \qquad \tilde{Y} = h_\phi^{(\mathrm{ref})}(M).$$

This SCM mirrors the wrapper SCM but holds $g_\theta^{(\mathrm{ref})}$ and $h_\phi^{(\mathrm{ref})}$ fixed so that causal effect estimates are computed under a stable model and are not biased by ongoing parameter updates. Following interventional mediation (Vansteelandt & VanderWeele, 2012) with multiple mediators (Vansteelandt & Daniel, 2017), we define the interventional indirect effect of concept $j$ by replacing $M_j$ with a matched counterfactual value while holding the remaining concept vectors fixed. Replacing all concept vectors yields the total effect $TE$, and the residual interaction satisfies

$$TE = \sum_{j=1}^{J} IE_j + IE_\mu,$$

without requiring any causal ordering or independence assumptions among mediators (Loh et al., 2022).

During training, these frozen snapshots are periodically refreshed (e.g. once per epoch), but each estimation step uses a fixed pair $(g_\theta^{(\mathrm{ref})}, h_\phi^{(\mathrm{ref})})$, ensuring that causal effects are always computed under a stable model.

Because $\tilde{Y}$ depends only on $M$, and $M$ depends only on $X$ through frozen deterministic maps (plus independent exogenous noise), the mediator-outcome ignorability condition $\tilde{Y}(m) \perp M \mid X$ holds *by construction*. This ensures that $IE_j, TE$, and $IE_\mu$ are identified within the causal target estimator SCM. A formal statement and proof appear in App. C.

## 5.2 CONCEPT LABELING AND COUNTERFACTUAL REFERENCE SETS

We obtain concept labels $c^{(\ell)}$ post hoc from state $X$ and use them both for concept supervision and to group logged samples into *concept reference sets*. For each concept $j$ and each label value $c_j^{(\ell)}$, we collect the tuples $(X_i, Z_i, M_i^{(\text{ref})}, \tilde{Y}_i)$ whose assigned label matches that value. These reference sets act as an empirical approximation to the conditional mediator distribution $P(M_j \mid X, c^{(\ell)})$.

To estimate the causal effect of concept $j$ for a query state $X$, we retrieve samples from the counterfactual reference set whose $X_i$ values are closest to the query in covariate space. We then form a *mediator hybrid* by replacing only the $j$th concept vector $M_j$ with the counterfactual reference value while holding all other concept vectors fixed. Evaluating the hybrid under the frozen modules $(g_\theta^{(\text{ref})}, h_\phi^{(\text{ref})})$ reveals how $\tilde{Y}$ changes when only concept $j$ is intervened upon (see App. E for details on handling mutually exclusive concepts). This contrastive procedure directly yields the interventional quantities $(IE_j, TE, IE_\mu)$ used for alignment. We report overlap diagnostics (coverage, ESS, fallback rate) to assess whether suitable reference samples exist near each query.[1]

## 5.3 ALGORITHM

CCW-Net's training proceeds in three phases (Alg. 1). Phase 1 estimates causal effects of each concept on actions from logged trajectories and concept labels. Phase 2 computes the wrapper policy's action sensitivity to concept changes. Phase 3 trains CCW-Net to imitate the expert policy, classify concepts, and align its concept-action sensitivity with Phase 1's causal estimates.

### 5.3.1 PHASE 1: ESTIMATE CAUSAL TRAINING TARGETS

We represent each logged sample as $(X, Z, M, \tilde{Y})$ and estimate interventional causal targets using the causal target estimator SCM. For each concept vector $M_j$, we define $IE_j$ by replacing $M_j$ while holding other concept vectors fixed; similarly, replacing all concept vectors yields $TE$, and $IE_\mu$ completes the decomposition $TE = \sum_j IE_j + IE_\mu$. Identification of these quantities requires mediator-outcome ignorability given $X$, positivity of the conditional mediator distribution $P(M \mid X)$,

---

**Algorithm 1: CCW-Net**

**Input:** Data $(X_i, c_i^{(\ell)}, Y_i)$; backbone $f$; encoder $g_\theta$; head $h_\phi$
**Init:** $g_\theta^{\text{ref}} \leftarrow g_\theta, h_\phi^{\text{ref}} \leftarrow h_\phi$. **foreach** *concept $j$ and label value $v$*
**do**
  build concept reference sets
  $\mathcal{R}_j[v] \leftarrow \{(X_i, g_\theta^{\text{ref}}(f(X_i))) : c_{ij}^{(\ell)} = v\}$ and compute standardization stats for $X$ in each set.
**for** *epoch* $= 1, \ldots, E$ **do**
  **if** *refresh time* **then**
    $g_\theta^{\text{ref}} \leftarrow g_\theta, h_\phi^{\text{ref}} \leftarrow h_\phi$;
    partially refresh each reference set $\mathcal{R}_j[v]$ and update its standardization stats.
  **for** *batch* $(X, c^{(\ell)}, Y)$ **do**
    `// Phase 1: Estimate causal targets`
    $(IE_j, TE, IE_\mu) \leftarrow$
      ESTIMATEEFFECTS$(X, c^{(\ell)}; g_\theta^{\text{ref}}, h_\phi^{\text{ref}})$;
    `// Phase 2: Compute wrapper`
    `   concept-action effects`
    $Z \leftarrow f(X), \quad M \leftarrow g_\theta(Z), \quad \hat{Y} \leftarrow h_\phi(M)$;
    **foreach** *active concept $j$* **do**
      $u_j \leftarrow M_j/(\|M_j\| + \varepsilon), \quad s_j \leftarrow J_{M_j}\hat{Y} u_j$;
    `// Phase 3: Causal alignment and`
    `   supervised losses`
    $\mathcal{L} \leftarrow \ell(\hat{Y}, Y) + \lambda_{\text{cls}} \cdot \text{CONCEPTLOSS}(M, c^{(\ell)}) + \lambda_{\text{align}} \cdot \sum_j \mathbb{1}_j (1 - \cos(s_j, IE_j))$;
    update $(\theta, \phi)$ by backprop on $\mathcal{L}$.

---

and consistency of the frozen snapshot. Unlike classical mediation, ignorability is satisfied *by construction* in CCW-Net's SCM (App. C.2).

To estimate these effects from logged trajectories, we draw nearest states from the appropriate concept reference set based on similarity in $X$, replace only the $j$th concept vector with the matched value, evaluate the resulting hybrids using $(g_\theta^{(\text{ref})}, h_\phi^{(\text{ref})})$, and average over $S$ Monte Carlo draws.

---

[1]These diagnostics quantify how often the estimator finds close matches in $X$. Without sufficient overlap, interventional effects cannot be reliably estimated.

Overlap diagnostics (coverage, ESS, fallback rate) assess the plausibility of the positivity condition (App. C.3).

### 5.3.2  PHASE 2: COMPUTE POLICY EFFECTS OF CONCEPTS ON ACTIONS

We obtain $Z = f(X)$, concept vectors $M = g_\theta(Z)$, and wrapper actions $\hat{Y} = h_\phi(M)$. For each concept $j$, we compute the local sensitivity

$$s_j = J_{M_j}\hat{Y}\, u_j, \qquad u_j = M_j/(\|M_j\| + \varepsilon),$$

which measures how the predicted action changes under an infinitesimal perturbation of concept $j$. This Jacobian-based formulation applies to any differentiable policy head.

### 5.3.3  PHASE 3: ALIGN POLICY EFFECTS TO CAUSAL TARGETS

CCW-Net aligns the model's local sensitivities $s_j$ with the interventional targets $IE_j$ using a masked cosine loss,

$$\mathcal{L}_{\text{align}} = \sum_j \mathbb{1}_j\big(1 - \cos(s_j, IE_j)\big),$$

applied only to active concepts. We optimize the full objective

$$\mathcal{L} = \mathcal{L}_{\text{imit}} + \lambda_{\text{cls}}\mathcal{L}_{\text{cls}} + \lambda_{\text{align}}\mathcal{L}_{\text{align}},$$

with alignment activated after a warm-up period. Reference sets are periodically refreshed to maintain support and reduce bias (App. C.3).

**Loss terms**  The imitation loss $\mathcal{L}_{\text{imit}}$ is the mean squared error between the wrapper action $\hat{Y}$ and the expert action $Y$. The concept classification loss $\mathcal{L}_{\text{cls}}$ applies the appropriate supervised objective for each architecture: cross-entropy for models whose concepts are predicted via logits (CBM and attention heads), and a margin loss on capsule lengths for capsule networks (Sabour et al., 2017) (App. F). The causal alignment loss $\mathcal{L}_{\text{align}}$ is the masked cosine penalty described above that encourages each sensitivity $s_j$ to match its corresponding interventional effect $IE_j$.

### 5.4  INTERPRETING THE LEARNED REPRESENTATION.

Although CCW-Net's causal targets are defined within the induced SCM and not the environment's dynamics, the aligned concept representation admits a useful interpretation in terms of local necessity and sufficiency for the model's action. For a given state $X$, the interventional effect $IE_j(X)$ quantifies how the action would change if only concept $j$ were counterfactually altered, while the total effect $TE(X)$ reflects the change from altering all concepts together. When $IE_j(X)$ matches the corresponding component of $TE(X)$—and other $IE_k$ and $IE_\mu$ are small—the action component is locally *sufficiently* generated by concept $j$ and is "owed" to the $X \to M_j \to \hat{Y}$ pathway in Pearl's sense. Conversely, if swapping $M_j$ removes that component, concept $j$ is locally *necessary* for it. The alignment loss therefore trains the concept vectors so that their directions encode these path-specific causal contributions in the latent space.

## 6  EMPIRICAL EVALUATION

We evaluate CCW-Net in a real-world two-aircraft aircraft formation task, extended trail (United States Air Force, 2024), in a high-fidelity F-16 physics environment (So & Fan, 2023; Heidlauf et al., 2018). This setting stresses causal, abstract, human-interpretable explanations. Identical pitch and roll commands can serve different concepts depending on scenario context (e.g. pursuit geometry and aircraft energy state) so explanations in action space are ambiguous while abstract concepts are informative.

### 6.1  AIRCRAFT FORMATION TASK

**Task**  We evaluate CCW-Net in simulation on a real-world, complex, underactuated task: an aircraft formation task called extended trail (United States Air Force, 2024; 2025), shown in Fig. 3,

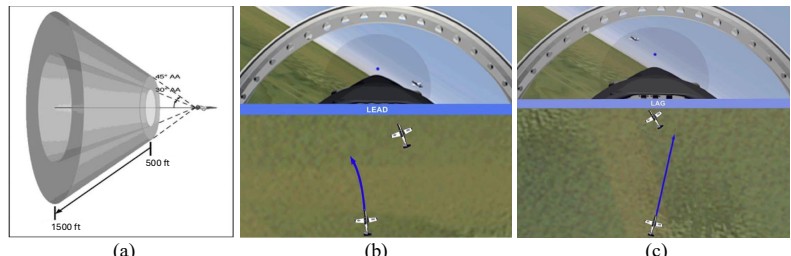

$$(a) \quad\quad\quad\quad\quad (b) \quad\quad\quad\quad\quad (c)$$

Figure 3: Extended trail task. (a) Extended trail cone. The chase aircraft (not depicted) is tasked with remaining within the dark shaded region of this cone while the lead aircraft (depicted) maneuvers (United States Air Force, 2024). (b) Lead pursuit. Performed by pointing the chase aircraft's nose ahead of the lead aircraft. Top: View from the chase aircraft cockpit. Bottom: Top-down view. (c) Lag pursuit. Performed by pointing the chase aircraft's nose behind the lead aircraft. Top: View from the chase aircraft cockpit. Bottom: Top-down view (United States Air Force, 2025).

where a chase aircraft maintains position behind a lead aircraft while executing dynamic maneuvers. With it comes four concepts that are used by real-world pilots to describe, perform, and debrief the task: Lead pursuit, lag pursuit, climb, and dive that hereby form the concept set $M = \{Lead, Lag, Climb, Dive\}$. Additional details are discussed in App. G.

**Environment**  A control loop simulates aircraft dynamics in three dimensions with a point-mass and six degrees of freedom (Heidlauf et al., 2018). Two aircraft are simulated. The lead aircraft follows a scripted path while the chase aircraft is trained to maintain formation with reinforcement learning (So & Fan, 2023) (see App. G.2 for details). The state includes relative kinematics (e.g., relative range, angle, and closure rates), ownship state, and normalized energy parameters. The action space is two dimensional in pitch and roll command.

**Expert policy and data**  A reinforcement-learning expert generates demonstration trajectories. We log tuples $(X, Z, Y)$ of raw observations $X$, frozen expert policy latents $Z$, and expert actions $Y$. Concept labels $c^{(\ell)} = (c_{LL}^{(\ell)}, c_{CD}^{(\ell)})$ corresponding to $\{Lead, Lag\} \times \{Climb, Dive\}$ are obtained via human- or auto-labeling (App. G.4). Within each $\{Lead, Lag\}$, $\{Climb, Dive\}$ concepts are physically mutually exclusive and therefore only one concept is active (i.e., an aircraft cannot both *Climb* and *Dive*, App. E.3).

**Why this task?**  Extended trail formation is a real-world, complex task in which pilots use the concepts $\{Lead, Lag, Climb, Dive\}$ to reason over, describe, and debrief with. These concepts are formally utilized in advanced flight training (United States Air Force, 2024; 2025) and serve to ground CCW-Net's utility for human-interpretability in complex real-world settings. From a robotics perspective, the task is not readily solved with inherently interpretable approaches. Furthermore, aircraft control, like many robotics applications, is underactuated making action-level labels poor proxies for human-meaningful concepts. The same action can implement different concepts depending on context. Additionally, sequences of actions are sometimes required to produce task-meaningful explanations.

## 6.2 Architectures Evaluated

Because $s_j$ is a directional derivative, CCW-Net applies to any differentiable head mapping concepts to actions. We instantiate three heads to demonstrate architectural generality.

All approaches adapt a key insight: expanding concept bottleneck models' (Koh et al., 2020) representations of concepts from scalars to vectors. In this vector representation, concept information is carried jointly through each vector's magnitude and direction, enabling richer, context-dependent structure than scalar concepts. We then supervise concepts with labeled data. This enables policies to reason over context-dependent representations of arbitrarily abstract, human-interpretable concepts. Notably, where scalar-based concept representations were adequate for the driving policies in

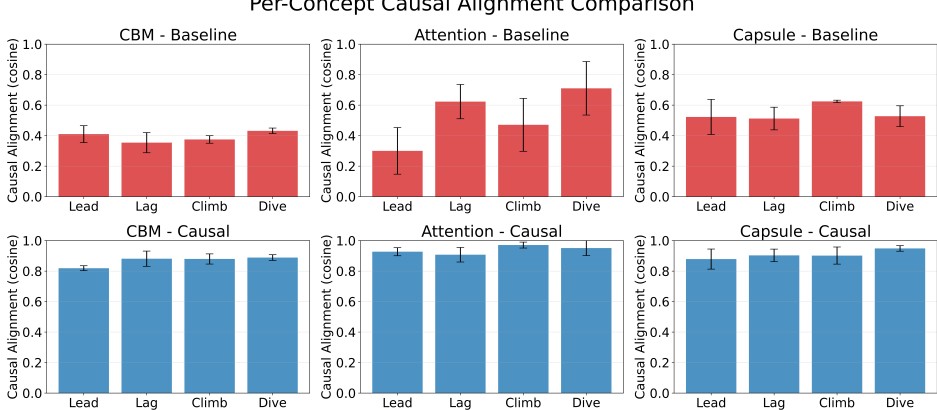

Figure 4: Causal Concept Alignment Before and After Causal Loss Applied. Red: CCW-Net Baseline with $\lambda_{\text{align}} = 0$. Blue: CCW-Net with $\lambda_{\text{align}} = 0.05$ applied. Causal alignment across all concepts and architectures is improved.

Kenny et al. (2023; 2024), they proved insufficient to reproduce the aircraft formation policy thereby motivating this work's extension to concept representations as vectors.

**Vector CBM**   We expand CBMs (Koh et al., 2020) to map $Z$ to vector-valued per-concept pre-activations, which are normalized to concept vectors $M_j$ (softmax over each block). Two-way concept classifiers (Lead/Lag, Climb/Dive) are supervised with cross-entropy on scores derived from these pre-activations. The action head is a per-concept linear map summed across concepts, followed by a $tanh$ nonlinearity.

**Capsule network with dynamic routing**   We introduce capsule networks' (Hinton et al., 2011) first known use in predicting actions for sequential decision making systems. Concepts are capsule vectors whose lengths encode activation and orientations encode expression. Dynamic routing connects concept capsules to action capsules based on context. Concepts are supervised through a margin loss (Sabour et al., 2017). Actions are transformed via $tanh$. We compute $s_j$ with Jacobians through the squash function, routing updates, and the final $tanh$.

**Attention head**   Each action dimension holds a learned query over concept key/values with learned $K, V$ projections. We compute $s_j$ via a Jacobian through a softmax attention. Across all heads, concepts are represented as vectors whose magnitude and direction together determine their contribution to the attention and action computations, enabling richer, human-interpretable abstractions.

## 7   RESULTS

**Hypotheses**   We test four hypotheses: **H1:** CCW-Net increases concepts' causal alignment with policy actions across all tested architectures; **H2:** CCW-Net does not qualitatively degrade main task performance relative to the frozen backbone; **H3:** CCW-Net does not degrade main task performance relative to a baseline wrapper with $\lambda_{\text{align}}=0$; and **H4** CCW-Net does not degrade concept-classification accuracy relative to that baseline.

**Results**   CCW-Net substantially improves mean causal alignment for all three heads: CBM $+0.449$ ($p < 0.01$), Capsule $+0.368$ ($p < 0.001$), and Attention $+0.234$ (not significant). We interpret the causal alignment for the attention head to be meaningful given that it achieves the greatest causal alignment score ($0.939\pm0.026$) (Fig. 4 and Table 4) thereby supporting **H1**. Test MSE remains comparable to the baseline wrapper (CBM $+0.002$, Capsule $+0.003$, Attention $+0.001$); the Capsule delta is statistically significant ($p < 0.05$), but its magnitude ($0.003$) is operationally negligible and was confirmed by visualizing rollouts, supporting **H3**. Qualitatively, 2-minute evaluation rollouts showed no extended trail cone violations when compared against the frozen backbone, supporting

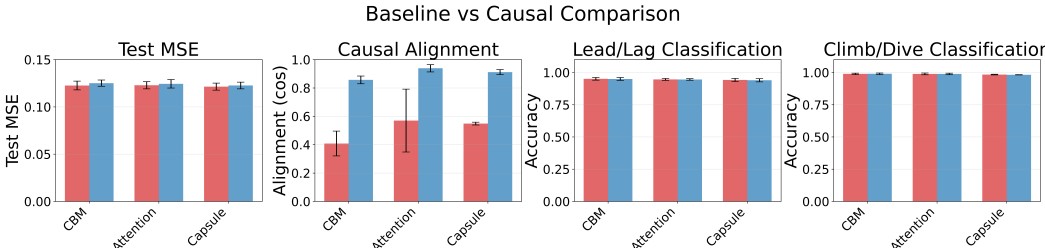

Figure 5: Change in Imitation MSE, Mean Causal Alignment, and Concept Classification Accuracy Due to Causal Losses. Red: CCW-Net Baseline with $\lambda_{\text{align}} = 0$. Blue: CCW-Net with $\lambda_{\text{align}} = 0.05$ applied. Causal alignment is improved while imitation and classification performance remain negligibly impacted.

**H2**. Concept classification was unchanged. Lead/Lag and Climb/Dive deltas are within 0.0-0.3 percentage points and not significant across heads (Fig. 5), supporting **H4**. Interestingly, training curves (Fig. 7) suggest causal effects quickly take hold in the policy.

**Summary**  CCW-Net consistently increases concept-to-action causal alignment (**H1**) without degrading task performance vs. the frozen backbone (**H2**) or baseline wrapper (**H3**), and without harming concept accuracy (**H4**). The effect holds across Capsule, vector CBM, and Cross-Attention heads, indicating CCW-Net is a practical, architecture-agnostic path to causally grounded, human-friendly explanations.

# 8 LIMITATIONS AND COMPUTATIONAL CONSIDERATIONS

CCW-Net's causal claims hold at the level of the induced SCM and logged expert data. It does not claim causal relationships with environment dynamics, which is left for future work. If concept labels or state coverage are biased or sparse, the estimated interventional effects $(IE_j, TE, IE_\mu)$ will reflect those limitations, and identification may fail when reference sets lack support. CCW-Net also assumes that the concept ontology provides sufficient support for the task where lack of coverage may be absorbed by the chosen concepts. Furthermore, CCW-Net assumes access to a sufficient quantity of labeled data to reproduce the frozen policy and achieve the desired level of causal alignment.

Computationally, CCW-Net adds a constant-factor overhead: beyond the usual forward pass, each batch requires one frozen-model forward plus roughly $JS$ additional frozen-head evaluations for mediator hybrids (where $J$ is the number of active concepts and $S$ is the number of Monte Carlo draws).

# 9 CONCLUSION

The world is moving fast toward deploying increasingly capable autonomous systems empowered by deep neural policies into high-stakes, real-world settings, but without transparency into *why* actions are taken, real use in sensitive domains remains limited. We introduced CCW-Net, a flexible wrapper that enables any differentiable policy head to reason over human-defined concepts and aligning them with *causal* effects estimated from observed trajectories. On a high-fidelity flight task, CCW-Net consistently improved concept-to-action causal alignment across three architectures, concept bottleneck models, attention heads, and capsule networks, while maintaining high task performance and concept classification accuracy. CCW-Net offers an architecture-agnostic, causally grounded approach to interpretability of deep neural policies. In this sense, CCW-Net provides a causally guided representation-learning objective, shaping the concept space so that its internal geometry reflects the estimated causal influence of each concept on the policy's decisions. In future work we look forward to extending causal connections to observations as well as producing causally-grounded counterfactual trajectories on the path toward enabling robots to answer "why?"

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

## A  VARIABLE DICTIONARY

| Symbol | Definition |
|---|---|
| $X$ | Raw observed features (covariates used for matching) |
| $Z$ | Frozen backbone latent $Z = f(X)$ |
| $c^{(\ell)}$ | Concept label(s) derived from $X$ and used only to index reference sets |
| $M = (M_1, \ldots, M_J)$ | Vector concept representations (per-concept normalized blocks) |
| $Y$ | Logged expert action (for imitation loss only) |
| $\hat{Y}$ | Wrapper action $h_\phi(M)$ |
| $\tilde{Y}$ | Reference action $h^{(\mathrm{ref})}(M^{(\mathrm{ref})})$ for target estimation |
| $M^{(\mathrm{ref})}$ | Concept vectors from the frozen encoder (reference mediators) |
| $g^{(\mathrm{ref})}, h^{(\mathrm{ref})}$ | Frozen snapshots used only in the target estimator SCM |
| $IE_j, TE, IE_\mu$ | Per-concept interventional effect, total effect, residual interaction |
| $u_j$ | Radial unit vector in block $j$, $u_j = M_j/(\|M_j\| + \varepsilon)$ |
| $s_j$ | Local concept $\rightarrow$ action sensitivity $s_j = J_{M_j}\hat{Y}\, u_j$ |
| $\mathcal{R}_j[c_j^{(\ell)}]$ | Reference set for concept $j$ and label value $c_j^{(\ell)}$, storing $(X_i, Z_i, M_i^{(\mathrm{ref})}, \tilde{Y}_i)$ |
| $D_i$ | Standardized squared distance in $X$ for matching |
| $w_i$ | Kernel weight for reference sample $i$ (Gaussian, bandwidth $\tau$) |
| $\delta, \tau, S$ | Caliper, kernel temperature, and Monte Carlo draws |
| $\mathcal{L}_{\mathrm{imit}}, \mathcal{L}_{\mathrm{cls}}, \mathcal{L}_{\mathrm{align}}$ | Imitation, concept supervision, and alignment losses |
| $\lambda_{\mathrm{cls}}, \lambda_{\mathrm{align}}$ | Loss weights for concept supervision and alignment |
| ESS, coverage, fallback | Standard overlap diagnostics for matching on $X$ |

Table 1: Variable dictionary.

## B  EXTENDED RELATED WORK

In this section, we discuss a few related research directions and explain how our approach differs from them.

**Causal attribution methods in deep learning**   Concept-based explanations have been adapted to control and planning including learning joint embeddings between state-action pairs and explanations (Das et al., 2023), gradient and path-based scored (Sundararajan et al., 2017), concept level testing (Kim et al., 2018), and self-explaining networks (Achtibat et al., 2023). Further work refines concept attribution and completeness guarantees (Yeh et al., 2020; Bai et al., 2023) and attributes relevance to produce human-understandable rationales (Achtibat et al., 2023). However, these approaches are correlational. In safety and task-critical autonomy, such correlational explanations can be misleading. Saliency or concept relevance can be high even when a factor is not on a causal path to the action or decision (Zhou et al., 2022). CCW-Net instead targets causal quantities. Our approach differs by explicitly grounding attribution in estimated causal effects. We not only compute how actions respond to concept changes, but also check these sensitivities against interventional baselines drawn from data.

**Interpretability of sequential decision making systems**   A large body of work endeavors to make sequential decision-making systems more interpretable (Sreedharan et al., 2022; Lanier et al., 2024; Li et al., 2025b; Verma & Shah, 2025). Post-hoc approaches explain behaviors via counterfactuals (Tsirtsis et al., 2021; Olson et al., 2021) and concept bottleneck models (Koh et al., 2020; Delfosse et al., 2024) to improve user understanding. In safety critical domains such as autonomous driving, surveys highlight gaps between correlational attributions and user needs for actionable, task-grounded explanations (Omeiza et al., 2021; Atakishiyev et al., 2025). The field further emphasizes balancing informativeness with cognitive load (Sanneman et al., 2024). CCW-Net advances this field by grounding concept-level explanations in causal effects on actions, rather than correlations, while providing information rich, real-world domain explanations.

**Causal explanations in sequential decision making**   Counterfactual reasoning is an accepted approach for explanations (Wachter et al., 2017) and has been explored for sequential decision making policies in multi-agent settings (Gyevnar et al., 2024), simple deterministic settings (Verma &

Srivastava, 2024), partially-observable Markov decision processes (POMDPs) through counterfactual information impact (Mahmud et al., 2024b), and recent surveys outlining causal explanation desiderata for sequential tasks (Nashed et al., 2025). Recent work also studies causally reliable concept bottlenecks (De Felice et al., 2025); however, our work goes beyond this by imposing causal relationships in within the policy.

**Causal explanations in reinforcement learning**   Causal approaches have been used to mitigate confounding in reinforcement learning (RL) (Li et al., 2025a), formulate causal imitation learning via inverse reinforcement learning (IRL) (Ruan et al., 2023), and to identify when causal reasoning benefits RL (Schulte & Poupart, 2025). Madumal et al. (Madumal et al., 2020) propose action-influence models that define an SCM over environment variables and agent actions and use it to produce contrastive explanations for RL policies. More recently, Kekić et al. (Kekić et al., 2025) learn nonlinear causal reductions that map a high-dimensional RL system to a lower-dimensional SCM whose interventional behavior matches the original policy. These methods operate at the environment level and do not impose a concept bottleneck or learn vector-valued mediators. CCW-Net contributes a complementary perspective by performing mediation analysis on demonstrations to infer concept-level causal targets and using them to shape the policy's internal concept-to-action representation, without requiring an explicit environment structural causal model.

**Discovering causal relationships in CBMs**   Recent work aims to reveal existing causal structure among concepts or endow concept models with causal meaning (Dominici et al., 2024; De Felice et al., 2025). Our work complements this such that instead of discovering a causal concept graph, we enhance the representation capacity of concept to carry both presence and context-dependent expression, and causally align concepts to actions by matching interventional causal targets.

**Causal abstractions of neural networks**   Work on causal abstractions studies when a neural network can implement a higher-level causal model and how to align abstract variables with internal representations (Geiger et al., 2021; 2025). These methods provide powerful tools for mechanistic analysis of classifiers and language models by validating an existing abstract SCM via interchange interventions. CCW-Net is complementary: rather than validating a fixed abstract model, we use interventional mediation effects to *train* a concept bottleneck representation inside a control policy so that per-concept directions in the latent space reflect their causal influence on actions.

## C   CAUSAL ASSUMPTIONS AND DIAGNOSTICS

### C.1   IDENTIFICATION ASSUMPTIONS FOR INTERVENTIONAL EFFECTS

CCW-Net computes per-concept interventional effects $IE_j$, the total effect $TE$, and the residual interaction $IE_\mu$ in the *causal target estimator SCM*

$$X \to Z \to M \to \tilde{Y},$$

where $Z = f(X)$, $M = g^{(\mathrm{ref})}(Z)$, and $\tilde{Y} = h^{(\mathrm{ref})}(M)$ (Fig. 2a). Following interventional multiple-mediator analysis (Vansteelandt & Daniel, 2017; Loh et al., 2022), these effects are identified under the following standard conditions:

- **Ignorability:** No unmeasured confounding of the mediator–outcome relation given $X$, i.e. $\tilde{Y}(m) \perp M \mid X$ in the causal target estimator SCM.
- **Positivity:** For all $x$ in the support of $X$, the conditional mediator distribution $P(M_j \mid X = x)$ has support in a neighborhood of the observed mediators. This ensures that the counterfactual mediator values required for swaps are well supported.
- **Consistency:** For the realized mediator value $M = m$, the observed $\tilde{Y}$ equals $\tilde{Y}(m)$ under the frozen decoder $h^{(\mathrm{ref})}$.

Under these conditions, the interventional effects satisfy the decomposition

$$TE \ = \ \sum_{j=1}^{J} IE_j \ + \ IE_\mu,$$

with $IE_\mu$ collecting all multi-mediator interaction terms that cannot be attributed to any single mediator (Loh et al., 2022).

## C.2 MODEL-LEVEL IGNORABILITY IN THE CAUSAL TARGET ESTIMATOR SCM

The ignorability condition above holds *by construction* in the causal target estimator SCM. During target estimation we treat the learned parameters of $f$, $g^{(\text{ref})}$, and $h^{(\text{ref})}$ as fixed constants, and assert mediator-outcome ignorability conditional on these parameters. The following proposition formalizes this in the presence of stochastic components and label noise.

**Proposition 1** (Model-level mediator–outcome ignorability)**.** *Consider the causal target estimator SCM with structural equations*

$$Z = f(X, U_f), \qquad M = g^{(\text{ref})}(Z, U_g), \qquad \tilde{Y} = h^{(\text{ref})}(M, U_h),$$

*where $U_f, U_g, U_h$ are exogenous noise variables. Suppose:*

*1. $U_h \perp (X, U_f, U_g)$ (exogenous independence of decoder noise);*

*2. $g^{(\text{ref})}$ and $h^{(\text{ref})}$ are frozen during target estimation;*

*3. All randomness used for matched reference sampling depends only on $(X, U_s)$ with $U_s \perp U_h$.*

*Then for every mediator value $m$, generic value of a concept vector $M_j$ under intervention, in the support of $M \mid X$,*

$$\tilde{Y}(m) \perp M \mid X.$$

*Hence the mediator–outcome ignorability required for identifying interventional effects $(IE_j, TE, IE_\mu)$ holds within the causal target estimator SCM.*

*Proof.* Expand the SCM to include noise terms. The parents of $\tilde{Y}(m, U_h)$ are only $(m, U_h)$, and by assumption $U_h$ is independent of $(X, U_f, U_g)$, and therefore of $(X, Z, M)$. Conditioning on $X$ blocks all back-door paths from $M$ to $\tilde{Y}(m, U_h)$ except those involving $U_h$, which are removed since $U_h$ is not a parent of $M$. Thus $\tilde{Y}(m, U_h) \perp M \mid X$, and averaging over $U_h$ yields $\tilde{Y}(m) \perp M \mid X$. Freezing $g^{(\text{ref})}$ and $h^{(\text{ref})}$ prevents training-induced dependencies, and independence of $U_s$ and $U_h$ ensures that matched reference sampling does not introduce additional links. □

**Practical enforcement** In practice, we evaluate $g^{(\text{ref})}$ and $h^{(\text{ref})}$ in deterministic (evaluation) mode and use independent random number generator (RNG) streams for sampling. This ensures that decoder noise cannot depend on the randomness that produced $M$, thereby satisfying the exogeneity requirement in Proposition 1.

**Noisy labels and stochastic modules** Concept labels $c^{(\ell)}$ are post-hoc functions of $X$ (possibly noisy). They are not part of the estimator SCM and therefore cannot confound $\tilde{Y}$ once we condition on $X$. If $f$, $g^{(\text{ref})}$, or $h^{(\text{ref})}$ include test-time stochasticity (e.g., dropout), it is absorbed into $(U_f, U_g, U_h)$. The key requirement is that $U_h$ is sampled independently from the randomness generating $M$, which is ensured by using evaluation mode or independent RNG streams.

## C.3 OVERLAP DIAGNOSTICS AS A PRACTICAL CHECK OF POSITIVITY

Positivity cannot be guaranteed in high-dimensional spaces, but practical violations can detected via standard overlap diagnostics computed during mediator matching. We use coverage, accepts-per-sample, effective sample size (ESS), standardized distance quantiles, and fallback rate to assess the quality of kernel matching on $X$. High coverage, low fallback, and large ESS indicate well-supported neighborhoods for interventional estimation. Table 2 reports these quantities across architectures and training regimes.

## C.4 INTERACTION TERM AND RELAXED MEDIATOR ASSUMPTIONS

Classical mediation analyses often assume a single mediator or strong restrictions (e.g., mediator independence). Interventional multiple-mediator effects do not require such assumptions. Instead, they decompose the total effect into

$$TE = \sum_j IE_j + IE_\mu,$$

where $IE_\mu$ collects all interactions and dependencies among mediators. This term is estimable in the causal target estimator SCM and is reported in Sec. 6. See Loh et al. (2022) for discussion.

## C.5 CONSEQUENCES OF ASSUMPTION VIOLATIONS

Identification errors in the causal target estimator SCM affect the quality of $IE_j$, $TE$, and $IE_\mu$ estimates. CCW-Net still aligns $s_j$ to these estimates, but the resulting explanations should be interpreted as causal *with respect to the induced SCM defined by the causal target estimator*, not the environment dynamics. Poor overlap or strong nonstationarities can increase bias or variance in the estimated causal targets—our overlap diagnostics are designed to surface such cases.

**Coverage.** Fraction of queries with at least one eligible neighbor under the caliper,

$$\text{coverage} \;=\; \frac{1}{B} \sum_{b=1}^{B} \mathbb{1}\Big[ \sum_i \mathbb{1}[D_{bi} \le \delta^2] > 0 \Big],$$

where $\mathbb{1}$ is the indicator function, $D_{bi}$ is the standardized squared distance from query $b$ to candidate $i$, $\delta$ is the caliper, and $B$ is the number of queries.

**Accepts per sample.** Mean number of eligible neighbors per query under the caliper,

$$\text{accepts/sample} \;=\; \frac{1}{B} \sum_{b=1}^{B} \sum_i \mathbb{1}[D_{bi} \le \delta^2].$$

**Effective sample size (ESS).** For kernel weights $w_{bi}$ normalized over eligible neighbors of query $b$,

$$\text{ESS} \;=\; \frac{1}{B} \sum_{b=1}^{B} \frac{1}{\sum_i w_{bi}^2}.$$

**Fallback rate.** Fraction of queries that required a nearest-neighbor fallback because no neighbor met the caliper,

$$\text{fallback rate} \;=\; \frac{\#\{\text{queries with zero eligible neighbors}\}}{B}.$$

**Distance quantiles.** Summary of standardized distances among eligible neighbors (min/median/90th percentile) to describe the tightness of the matching neighborhood in $X$ (covariate space).

High coverage and low fallback indicate adequate support in $X$ for the counterfactual draws used in interventional estimation. Larger accepts/sample and ESS reflect richer local neighborhoods and lower variance in the Monte-Carlo estimates. These diagnostics, along with Table 2, provide empirical evidence that positivity holds to a reasonable degree in our experiments.

Table 2: Overlap diagnostics by architecture and training regime (baseline vs. causal).

| Architecture | Regime | Accepts/sample | Coverage | ESS | Fallback rate (%) |
|---|---|---|---|---|---|
| CBM | Baseline | 0.426±0.004 | 0.984±0.002 | 230.270±3.980 | 0.031±0.004 |
| CBM | Causal | 0.428±0.006 | 0.986±0.002 | 231.356±8.288 | 0.028±0.004 |
| Capsule | Baseline | 0.427±0.003 | 0.985±0.002 | 229.222±3.507 | 0.030±0.003 |
| Capsule | Causal | 0.428±0.006 | 0.986±0.002 | 231.356±8.288 | 0.028±0.004 |
| Attention | Baseline | 0.427±0.003 | 0.985±0.002 | 229.744±4.359 | 0.031±0.004 |
| Attention | Causal | 0.428±0.006 | 0.986±0.002 | 231.356±8.288 | 0.028±0.004 |

*Note:* Accepts/sample = mean eligible neighbors under the caliper; Coverage = fraction with at least one eligible neighbor; ESS = effective sample size $1/\sum_i w_i^2$ averaged over queries; Fallback rate = fraction requiring nearest-neighbor fallback because no eligible neighbors were found under the caliper. Values are mean $\pm$ 95% CI across runs.

**Interpretation**  High coverage and low fallback indicate that almost all queries have at least one nearby reference concept in their treatment group under the caliper, supporting practical positivity. Larger accepts/sample and ESS values reflect richer local neighborhoods and lower variance in the Monte-Carlo effect estimates. Distance quantiles summarize how tight these neighborhoods are in $X$: smaller values correspond to closer matches. In Table 2, coverage $\approx 0.98\text{--}0.99$, ESS $\approx 230$, and fallback $\approx 0.03\%$ together suggest that overlap is good and our interventional estimates are based on well-supported, low-variance neighborhoods in covariate space.

## C.6   Interaction term and relaxed mediator assumptions

Classical mediation decompositions often assume either a single mediator or strong restrictions on the mediator graph (e.g., no interactions, fixed ordering). Following Loh et al. (2022), we instead decompose the total effect as

$$TE = \sum_j IE_j + IE_\mu,$$

where $IE_j$ are per-mediator interventional effects and $IE_\mu$ is a residual interaction term. This allows for context-dependent interactions among mediators without specifying a causal ordering or exclusion restrictions among the $M_j$, and collects any multi-mediator effects into $IE_\mu$.

### C.6.1   Residual interaction share

We report the residual interaction share

$$\kappa_\mu = \frac{\langle IE_\mu, TE \rangle}{\|TE\|^2}$$

per architecture, with and without causal losses applied. $\kappa_\mu$ represents the fraction of the total effect explained by interactions between mediators rather than by any single concept alone; it can be negative when interactions oppose the total effect.

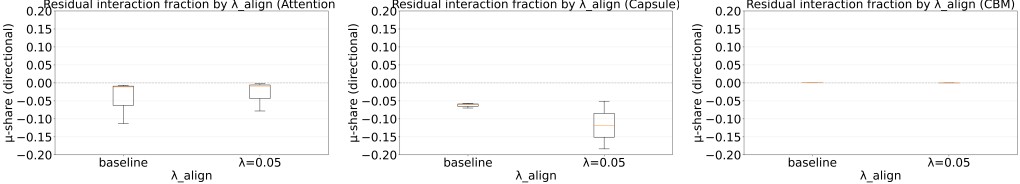

Figure 6: Residual interaction share across architectures with and without causal loss applied. Negative values imply effects against the total effect.

## C.7   Consequences of assumption violations

Our method has two stages: (i) estimate interventional effects $(IE_j, TE, IE_\mu)$ from logged data, and (ii) align the wrapper's local sensitivities $s_j$ to these targets. If the identification assumptions

hold, the targets correspond to the true interventional effects in the causal target estimator SCM, and CCW-Net produces concept-attribution that matches those effects.

If the assumptions hold only approximately, the estimated $IE_j$ are biased relative to the true SCM, and CCW-Net aligns to these approximate interventional effects. The optimization still runs, but the causal interpretation of the explanations becomes "with respect to the induced SCM and covariate set we use," rather than with respect to the ground-truth environment. Severe positivity violations (e.g., low coverage or high fallback in certain regions) also increase variance and potential bias in the estimated $IE_j$; in such regions, the learned alignment is correspondingly less trustworthy. Our overlap diagnostics and interaction-share analyses are intended to surface such pathologies in practice.

# D   CCW-NET TRAINING

## D.1   TRAINING SUMMARY

We train CCW-Net on $500{,}000$ expert state-action samples with concept labels. Labeled samples (with concept pairs) are split into train/val/test (train $60\%$, test $15\%$, validation $15\%$) with stratification over the joint concept label. The pretrained backbone $f$ is frozen and we train only the concept encoder $g_\theta$ and the policy head $h_\phi$. Training runs for $100$ epochs. A warm-up of $50$ epochs uses imitation and concept supervision and the causal alignment loss is activated after warm-up. We use the Adam optimizer (learning rate $1 \times 10^{-4}$), batch size $256$, and five random seeds per setting. We minimize

$$\mathcal{L} = \mathcal{L}_{\text{imit}} + \lambda_{\text{cls}}\, \mathcal{L}_{\text{cls}} + \lambda_{\text{align}}\, \mathcal{L}_{\text{align}},$$

with $\lambda_{\text{align}} = 0$ during warm-up. Concept supervision uses a margin loss on capsule lengths for the capsule network (App. F.1), and two-way cross-entropy on per-concept scores derived from pre-activations for the CBM and attention heads (Apps. F.2 and F.3). In all cases, the supervised quantities are deterministic functions of the concept encoder outputs that also parameterize the mediators $M_j$ used for actions.

**Interventional estimation (Monte Carlo)**  Treatment groups are the concept-pair bundles $(L, C) \in \{\text{Lead,Lag}\} \times \{\text{Climb,Dive}\}$. Per group we store $(X_i, M_i)$ and standardization $(\mu, \sigma)$. Matching uses standardized squared distance with caliper $\delta = 1.5$ and Gaussian-kernel weights with temperature $\tau = 0.25$; reference concepts are sampled via the Gumbel-Max trick with nearest-neighbor fallback. We use $S = 8$ draws per swap and compute $IE_j$, $TE$, and $IE_\mu$ with the frozen decoder saved in the reference sets. Reference sets are cross-fitted (two folds), first built after epoch 2 (given warm-up $\geq 10$ epochs), and then refreshed every epoch via partial replacement fraction $\rho = 0.1$ to reduce distributional drift.

**Diagnostics and reporting**  We report test MSE, concept accuracies, mean active alignment $\cos(s_j, IE_j)$, per-concept cosine, and the signed interaction share $\kappa_\mu = \langle IE_\mu, TE \rangle / \|TE\|^2$. Overlap diagnostics for matching include accepts/sample, coverage, effective sample size (ESS), standardized distance quantiles, and fallback rate.

Table 3: CCW-Net training settings (shared across architectures unless noted).

| Item | Setting |
|---|---|
| Training set size | 500,000 samples |
| Epochs / Warm-up | 100 / 50 (alignment off during warm-up) |
| Batch size / Optimizer / LR | 256 / Adam / $1 \times 10^{-4}$ |
| Seeds per setting | 5 |
| Frozen vs. trained | Backbone frozen; train $g_\theta$ and $h_\phi$ |
| Concept supervision | Capsules: margin ($m_+ = 0.9$, $m_- = 0.1$); CBM/Attn: cross-entropy per pair |
| Alignment loss | Masked cosine on active concepts only |
| Jacobian | $s_j = J_{M_j} \hat{Y} u_j$, $u_j = M_j/(\|M_j\| + \varepsilon)$ (one directional derivative per concept) |
| MC draws / Matching | $S = 8$; caliper $\delta = 1.5$, kernel $\tau = 0.25$, Gumbel-Max sampling |
| Reference refresh | Cross-fitted; partial replacement $\rho = 0.1$ each refresh (every epoch) |
| Checkpoints | Every 5 epochs; final at epoch 100 |
| Reported metrics | Test MSE, Acc-LL, Acc-CD, mean active cosine, per-concept cosine, $\kappa_\mu$ |

## D.2 TRAINING CURVES

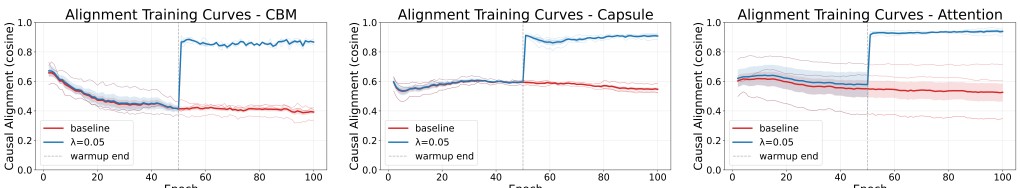

Figure 7: Causal training curves for the concept bottleneck model (CBM), capsule network, and attention architectures. Causal losses applied at epoch 50.

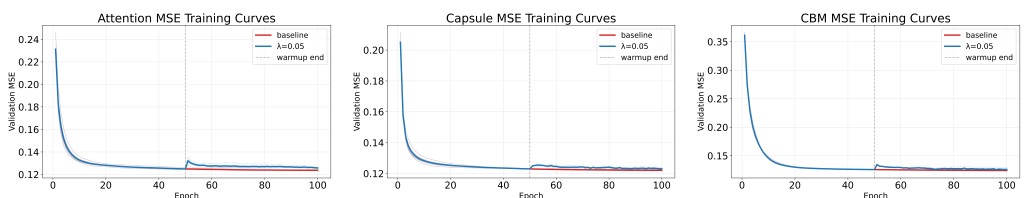

Figure 8: Imitation training curves (mean squared error - MSE) for the concept bottleneck model (CBM), capsule network, and attention architectures. Causal losses applied at epoch 50.

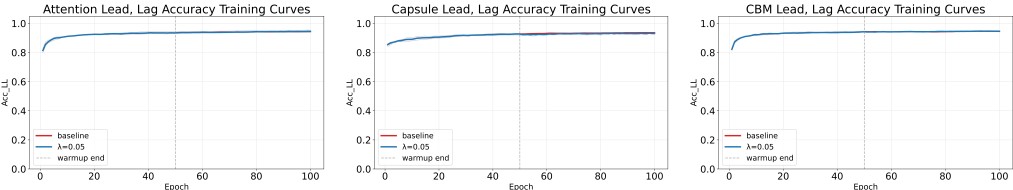

Figure 9: Lead/Lag pursuit concept classification training curves for the concept bottleneck model (CBM), capsule network, and attention architectures. Causal losses applied at epoch 50.

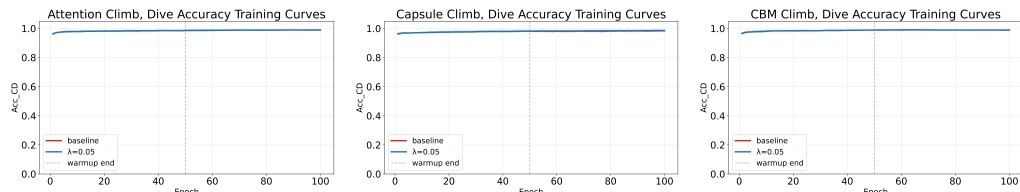

Figure 10: Climb/Dive concept classification training curves for the concept bottleneck model (CBM), capsule network, and attention architectures. Causal losses applied at epoch 50.

### D.3 DATA SUMMARY

#### D.3.1 STATISTICAL SUMMARY

Table 4: Performance and causal alignment by architecture at $\lambda_{\text{align}} = 0.05$ (mean±95% CI). $\Delta$ rows report Causal−Baseline with significance from paired tests.

| Architecture | Regime | Test MSE ↓ | Acc_LL ↑ | Acc_CD ↑ | Mean Causal Alignment ↑ |
|---|---|---|---|---|---|
| CBM | Baseline | 0.122±0.003 | 0.950±0.006 | 0.992±0.005 | 0.356±0.044 |
| | Causal ($\lambda$=0.05) | 0.125±0.002 | 0.949±0.007 | 0.992±0.004 | 0.855±0.018 |
| | $\Delta$ (C−B) | +0.003 ** | −0.001 ns | +0.000 ns | +0.500 *** |
| Capsule | Baseline | 0.121±0.002 | 0.944±0.007 | 0.987±0.004 | 0.555±0.004 |
| | Causal ($\lambda$=0.05) | 0.122±0.002 | 0.938±0.007 | 0.985±0.003 | 0.905±0.013 |
| | $\Delta$ (C−B) | +0.001 ** | −0.006 ns | −0.002 ns | +0.350 *** |
| Attention | Baseline | 0.123±0.002 | 0.944±0.006 | 0.991±0.005 | 0.404±0.144 |
| | Causal ($\lambda$=0.05) | 0.124±0.003 | 0.944±0.005 | 0.991±0.004 | 0.938±0.015 |
| | $\Delta$ (C−B) | +0.002 * | +0.000 ns | −0.000 ns | +0.534 * |

*Note:* Significance levels: *** $p < 0.001$, ** $p < 0.01$, * $p < 0.05$, ns = not significant. Acc_LL = Lead/Lag accuracy, Acc_CD = Climb/Dive accuracy, Mean Causal Alignment = mean active cosine. Causal alignment is cosine similarity in $[-1, 1]$. Accuracies are proportions in $[0, 1]$. Values are mean $\pm$ 95% CI across seeds ($n = 5$). All tests use paired samples across matched seeds.

#### D.3.2 TRAIN−VAL−TEST OBJECTIVE SUMMARY

We report train, validation, and test performance for imitation error, concept classification, and causal alignment across architectures and regimes (Tables 5–8). These metrics complement the test-only statistical comparisons in Table 4 and demonstrate minimal train–test gaps.

**Train−Val−Test Imitation Error**

Table 5: Imitation performance (MSE; mean±95 percent CI over 5 seeds). Baseline corresponds to $\lambda_{\text{align}} = 0$ and Causal corresponds to $\lambda_{\text{align}} = 0.05$.

| Architecture | Regime | Train MSE ↓ | Val MSE ↓ | Test MSE ↓ |
|---|---|---|---|---|
| CBM | Baseline | $0.1203 \pm 0.0002$ | $0.1214 \pm 0.0034$ | $0.1223 \pm 0.0027$ |
| CBM | Causal | $0.1228 \pm 0.0004$ | $0.1239 \pm 0.0034$ | $0.1253 \pm 0.0019$ |
| Attention | Baseline | $0.1202 \pm 0.0002$ | $0.1214 \pm 0.0028$ | $0.1227 \pm 0.0022$ |
| Attention | Causal | $0.1222 \pm 0.0004$ | $0.1229 \pm 0.0034$ | $0.1242 \pm 0.0028$ |
| Capsule | Baseline | $0.1193 \pm 0.0001$ | $0.1198 \pm 0.0028$ | $0.1212 \pm 0.0022$ |
| Capsule | Causal | $0.1204 \pm 0.0002$ | $0.1209 \pm 0.0028$ | $0.1224 \pm 0.0020$ |

**Train−Val−Test Concept Classification**

Table 6: Lead/Lag classification accuracy (Acc_LL; mean±95% CI over 5 seeds). Baseline corresponds to $\lambda_{\text{align}} = 0$ and Causal corresponds to $\lambda_{\text{align}} = 0.05$.

| Architecture | Regime | Train Acc_LL ↑ | Val Acc_LL ↑ | Test Acc_LL ↑ |
|---|---|---|---|---|
| CBM | Baseline | $0.948 \pm 0.001$ | $0.947 \pm 0.003$ | $0.950 \pm 0.006$ |
| CBM | Causal | $0.948 \pm 0.001$ | $0.946 \pm 0.003$ | $0.949 \pm 0.007$ |
| Attention | Baseline | $0.945 \pm 0.001$ | $0.943 \pm 0.006$ | $0.944 \pm 0.006$ |
| Attention | Causal | $0.945 \pm 0.001$ | $0.943 \pm 0.006$ | $0.944 \pm 0.005$ |
| Capsule | Baseline | $0.938 \pm 0.002$ | $0.936 \pm 0.003$ | $0.944 \pm 0.007$ |
| Capsule | Causal | $0.933 \pm 0.002$ | $0.932 \pm 0.005$ | $0.938 \pm 0.007$ |

Table 7: Climb/Dive classification accuracy (Acc_CD; mean±95% CI over 5 seeds). Baseline corresponds to $\lambda_{\text{align}} = 0$ and Causal corresponds to $\lambda_{\text{align}} = 0.05$.

| Architecture | Regime | Train Acc_CD ↑ | Val Acc_CD ↑ | Test Acc_CD ↑ |
|---|---|---|---|---|
| CBM | Baseline | $0.992 \pm 0.000$ | $0.988 \pm 0.002$ | $0.992 \pm 0.005$ |
| CBM | Causal | $0.991 \pm 0.000$ | $0.989 \pm 0.003$ | $0.992 \pm 0.004$ |
| Attention | Baseline | $0.991 \pm 0.000$ | $0.988 \pm 0.003$ | $0.991 \pm 0.005$ |
| Attention | Causal | $0.991 \pm 0.000$ | $0.988 \pm 0.004$ | $0.991 \pm 0.004$ |
| Capsule | Baseline | $0.987 \pm 0.001$ | $0.985 \pm 0.004$ | $0.987 \pm 0.004$ |
| Capsule | Causal | $0.985 \pm 0.001$ | $0.983 \pm 0.004$ | $0.985 \pm 0.003$ |

**Train–Val–Test Causal Alignment**

Table 8: Causal alignment (mean active cosine; mean±95% CI over 5 seeds). Only causal runs use the alignment objective; for all results shown, $\lambda_{\text{align}} = 0.05$. Baseline runs have no alignment objective and are therefore not listed.

| Architecture | Train Align ↑ | Val Align ↑ | Test Align ↑ |
|---|---|---|---|
| CBM | $0.851 \pm 0.005$ | $0.858 \pm 0.020$ | $0.855 \pm 0.018$ |
| Attention | $0.937 \pm 0.015$ | $0.933 \pm 0.018$ | $0.938 \pm 0.015$ |
| Capsule | $0.907 \pm 0.010$ | $0.904 \pm 0.014$ | $0.905 \pm 0.013$ |

# E  INTERVENTIONAL ESTIMATION AND ALIGNMENT DETAILS

## E.1  NOTATION AND SETUP

Each logged sample is represented as $(X, Z, M, \tilde{Y})$ in the causal target estimator SCM, where

$$Z = f(X), \qquad M = g_\theta^{(\text{ref})}(Z), \qquad \tilde{Y} = h_\phi^{(\text{ref})}(M).$$

Concept labels $c^{(\ell)}$ are post hoc functions of $X$ and are used only to form *concept reference sets*. For each concept $j$ and each label value of $c_j^{(\ell)}$, we define a reference set

$$\mathcal{R}_j[c_j^{(\ell)}] = \{(X_i, Z_i, M_i^{(\text{ref})}, \tilde{Y}_i) : c_{ij}^{(\ell)} = c_j^{(\ell)}\},$$

together with per-set standardization statistics for $X$. These sets approximate the conditional distribution $P(M_j \mid X, c_j^{(\ell)})$, where labels partition samples by concept value and matching is performed in $X$-space.

## E.2  MATCHING AND WEIGHTED SAMPLING

Given a query state $X^*$ and concept label $c_j^{(\ell)*}$, we match among states in $\mathcal{R}_j[c_j^{(\ell)*}]$. Let $\mu_j$ and $\sigma_j$ denote the mean and standard deviation of $X$ in this reference set. We standardize

$$\tilde{X}_i = \frac{X_i - \mu_j}{\sigma_j}, \qquad \tilde{X}^* = \frac{X^* - \mu_j}{\sigma_j},$$

and calculate standardized squared distances

$$D_i = \frac{\|\tilde{X}^* - \tilde{X}_i\|_2^2}{d_X}.$$

Candidates with $D_i \leq \delta^2$ form the eligible set. Gaussian kernel weights are assigned as

$$w_i \propto \exp\left(-\frac{D_i}{2\tau^2}\right), \qquad \sum_i w_i = 1.$$

If no candidate satisfies the caliper, we fall back to the nearest neighbor. We report coverage, accepts-per-sample, ESS, and fallback rate in App. C.3.

### E.3 MONTE CARLO MEDIATOR REPLACEMENTS

For each active concept $j$, we draw $S$ weighted samples from $\mathcal{R}_j[c_j^{(\ell)*}]$ using a Gumbel–Max sampler over $\{w_i\}$. For each draw we construct a mediator hybrid

$$M_{\text{hyb}(j)}^{(s)} = (M_1, \ldots, M_{j-1}, M_j^{(\text{ref})(s)}, M_{j+1}, \ldots, M_J),$$

and evaluate it under the frozen decoder

$$\tilde{Y}_{\text{hyb}(j)}^{(s)} = h_\phi^{(\text{ref})}\big(M_{\text{hyb}(j)}^{(s)}\big).$$

**Handling Mutually Exclusive Concepts** In some domains, including the aircraft task studied here, certain concepts occur in mutually exclusive pairs (e.g., left vs right turn). Our block-wise hybrid construction naturally handles both mutually exclusive concepts (where swapping $M_j$ implicitly toggles between the two alternatives, since each reference set corresponds to one member of the mutually exclusive pair) and presence-based concepts (where $M_j$ is replaced by an 'absent' baseline).

For the total effect, all active concept blocks are replaced jointly:

$$M_{\text{hyb(all)}}^{(s)} \quad \text{and} \quad \tilde{Y}_{\text{hyb(all)}}^{(s)} = h_\phi^{(\text{ref})}(M_{\text{hyb(all)}}^{(s)}).$$

### E.4 EFFECT DECOMPOSITION

Let $\tilde{Y}_{\text{obs}} = h_\phi^{(\text{ref})}(M)$ denote the reference output for the observed mediator. Monte Carlo estimates of the interventional effects are

$$IE_j = \frac{1}{S} \sum_{s=1}^{S} \left(\tilde{Y}_{\text{obs}} - \tilde{Y}_{\text{hyb}(j)}^{(s)}\right),$$

$$TE = \frac{1}{S} \sum_{s=1}^{S} \left(\tilde{Y}_{\text{obs}} - \tilde{Y}_{\text{hyb(all)}}^{(s)}\right),$$

$$IE_\mu = TE - \sum_j IE_j.$$

This follows the interventional multiple-mediator decomposition of Loh et al. (2022), where interaction terms capture dependence among mediators.

### E.5 LOCAL POLICY EFFECTS AND ALIGNMENT

For wrapper outputs $\hat{Y} = h_\phi(M)$, the local sensitivity of actions to concept $j$ is

$$s_j = J_{M_j}\hat{Y}\, u_j, \qquad u_j = \frac{M_j}{\|M_j\| + \varepsilon}.$$

We align these sensitivities to the interventional targets using the masked cosine loss

$$\mathcal{L}_{\text{align}} = \sum_j \mathbb{1}_j (1 - \cos(s_j, IE_j)),$$

and optimize the total training objective

$$\mathcal{L} = \mathcal{L}_{\text{imit}} + \lambda_{\text{cls}}\mathcal{L}_{\text{cls}} + \lambda_{\text{align}}\mathcal{L}_{\text{align}}.$$

### E.6 IMPLEMENTATION SUMMARY

For each minibatch:

1. Compute $Z = f(X)$, $M = g_\theta(Z)$, and $\hat{Y} = h_\phi(M)$.

2. Compute $s_j$ for all active concepts.

3. Draw weighted reference samples using the appropriate concept label $c^{(\ell)}$ and matching in $X$.

4. Form mediator hybrids, evaluate $\tilde{Y}$ under $h_\phi^{(\text{ref})}$, and estimate $IE_j$, $TE$, and $IE_\mu$.

5. Compute losses and update $(\theta, \phi)$.

6. Periodically refresh frozen snapshots and update reference sets.

## F POLICY HEADS EVALUATED FOR CCW-NET

**High-level architecture** The concept encoder produces per-concept pre-activations $r_j \in \mathbb{R}^{d_j}$, which each architecture maps to concept vectors $M_j$ using its own normalization method (e.g., capsule squash, softmax, or $\tanh$). Collecting $M = (M_1, \ldots, M_J)$, the policy head maps

$$h_\phi : \mathbb{R}^{\sum_j d_j} \to \mathbb{R}^{|Y|}, \qquad \hat{Y} = h_\phi(M).$$

Causal alignment uses the local per-concept sensitivity

$$s_j = J_{M_j} \hat{Y} \, u_j, \qquad u_j = \frac{M_j}{\|M_j\| + \varepsilon},$$

the directional derivative of the wrapper action with respect to concept vector $M_j$. For architectures where $M_j$ is obtained from pre-activations $r_j$ via a nonlinear normalization (e.g., softmax or $\tanh$), $s_j$ is computed using the chain rule through that normalization, as detailed in the following subsections.

### F.1 CAPSULE NETWORK WITH DYNAMIC ROUTING

Capsule networks instantiate concepts as vectors whose length represents presence or activation and whose orientation captures context-dependent expression.

**Representation** Given pre-activations $r \in \mathbb{R}^{J \times d}$, each concept block is normalized via the capsule "squash" nonlinearity

$$M_j = \frac{\|r_j\|^2}{1 + \|r_j\|^2} \frac{r_j}{\|r_j\|},$$

so $\|M_j\| \in (0, 1)$ encodes presence and direction encodes expression.

**Action mapping with routing** Votes $u_{jk} = W_{jk} M_j$ are routed to $K$ action capsules using agreement-refined coefficients $c_{jk}$:

$$s_k = \sum_j c_{jk} \, u_{jk}, \qquad v_k = \text{squash}(s_k).$$

The action is

$$\hat{Y} = h_\phi(M) = \tanh\left(\frac{1}{K} \sum_{k=1}^{K} P_k v_k\right), \quad P_k \in \mathbb{R}^{d \times |Y|}.$$

**Concept supervision** Margin loss is applied to capsule lengths $\|M_j\|$ for labeled concept pairs, so supervision directly encourages length to encode presence.

**Attribution for alignment** For this head, the sensitivity $s_j$ is implemented by chaining Jacobians through the squash nonlinearity, the stored routing updates, the linear action maps, and the final $\tanh$, matching the implementation in the training code.

### F.2 VECTOR CONCEPT BOTTLENECK MODEL (CBM)

Here, we extend typical scalar CBMs to vector-valued concepts.

**Representation** Given pre-activations $r \in \mathbb{R}^{J \times d}$, each concept vector is block-normalized by a softmax applied within each concept block:

$$M_j = \psi_j(r_j) = \mathrm{softmax}(r_j).$$

**Action mapping** A per-concept linear action is summed and passed through $\tanh$:

$$\hat{Y} = h_\phi(M) = \tanh\Big(\sum_{j=1}^{J} P_j M_j + b\Big), \quad P_j \in \mathbb{R}^{d \times |Y|},\ b \in \mathbb{R}^{|Y|}.$$

**Concept supervision** For each labeled concept pair, two logits are formed from block-means of the pre-activations $r$ and trained with standard two-way cross-entropy (Lead vs. Lag, Climb vs. Dive). Thus supervision operates on scores derived from $r_j$, which deterministically parameterize the concept vectors $M_j$.

**Attribution for alignment** With a radial direction in pre-activation space $u_j^r = r_j/(\|r_j\| + \varepsilon)$, the local effect can be written as

$$s_j = J_{r_j}(h_\phi \circ \psi)(r)\, u_j^r = J_{M_j} h_\phi(M)\, u_j^M, \quad u_j^M = J_{r_j} \psi_j(r_j)\, u_j^r,$$

which corresponds to the softmax Jacobian composed with the linear output and the final $\tanh$. In practice we compute $s_j$ via this chain rule through $\psi_j$ and $h_\phi$.

### F.3 CROSS-ATTENTION

**Representation** Given pre-activations $r \in \mathbb{R}^{J \times d}$, a non-capsule normalization yields

$$M_j = \tanh(r_j).$$

**Action mapping via attention** Each action dimension $y \in \{1, \ldots, |Y|\}$ attends to concepts using a learned query $q_y \in \mathbb{R}^d$ and key/value projections

$$k_j = K_{\mathrm{proj}} M_j, \qquad v_j^{\mathrm{val}} = V_{\mathrm{proj}} M_j, \qquad \alpha_{yj} = \mathrm{softmax}_j\Big(\frac{q_y^\top k_j}{\sqrt{d}}\Big).$$

With context vectors $\mathrm{ctx}_y = \sum_j \alpha_{yj} v_j^{\mathrm{val}}$, the action is

$$\hat{Y} = h_\phi(M) = \tanh\big(W_{\mathrm{out}}\, \mathrm{ctx} + b\big), \quad W_{\mathrm{out}} \in \mathbb{R}^{|Y| \times d},\ b \in \mathbb{R}^{|Y|}.$$

**Concept supervision** As in the CBM head, two logits per labeled concept pair are derived from block-means of $r$ and trained with cross-entropy, so supervision operates on scores derived from the same pre-activations that parameterize $M_j$.

**Attribution for alignment** For this head, $s_j$ is implemented via a single Jacobian–vector product through the attention softmax and the final $\tanh$, using the radial direction $u_j = M_j/(\|M_j\| + \varepsilon)$ in concept space.

## G ADDITIONAL DOMAIN INFORMATION

### G.1 AIRCRAFT FORMATION TASK

Extended trail formation flight is a real-world, complex task used in formal advanced pilot training to teach pilots how to manage aircraft position and energy with respect to another aircraft. Lead pursuit, lag pursuit, climb, and dive (represented as $\{Lead, Lag, Climb, Dive\}$ concepts in CCW-Net) are well grounded in human-interpretability as they are formally used to teach, communicate, fly, and debrief the extended trail formation task.

### G.1.1 EXTENDED TRAIL TASK

Extended trail consists of a formation of two aircraft: a lead and a chase aircraft. The chase aircraft's task is to remain within a defined cone behind the lead aircraft. The extended trail cone is defined as $30°$ to $45°$ off of the lead aircraft's tail and between 500 feet and 1500 feet behind the lead aircraft (Fig. 3).

## G.2 POLICY TRAINING

The expert policy is trained using the method from So & Fan (2023), a variant of PPO (Schulman et al., 2017) that additionally considers per-timestep safety constraints. The original aircraft environment (Heidlauf et al., 2018; So & Fan, 2023) with a 4-dimensional control space consisting of the desired load factor, desired roll rate, desired yaw rate, and throttle, and a 20-dimensional observation space consisting of the states of the two aircraft and the current control. During training, we fix the throttle and set the desired yaw rate to 0. Moreover, to improve the temporal smoothness of the controls, we control the change in the control outputs and store the current controls in the state. The training framework is implemented in JAX (Bradbury et al., 2018) in the JAX version of the aircraft environment (So & Fan, 2023).

## G.3 EXTENDED TRAIL CONCEPTS

Four concepts are used to teach, communicate, fly, and debrief the extended trail task: Lead pursuit, lag pursuit, climb, and dive (United States Air Force, 2025) (Fig. 3). Lead and lag pursuit are fundamental maneuvers and application to flight throughout aviation. Lead pursuit is used to catch up with the lead aircraft while lag pursuit is used to increase distance from the lead aircraft. Lead pursuit is performed by pointing the nose of the chase aircraft ahead of the lead aircraft. Conversely, lag pursuit is performed by pointing the nose of the chase aircraft behind the lead aircraft. Climb and dive are conceptually simpler in that they are performed by ascending or descending in altitude.

There are infinitely many ways to perform lead pursuit, lag pursuit, climb and dive in two-dimensional action space (pitch and roll). For example, a dive may be performed while aircraft is at any roll angle (e.g., upright, inverted, or any angle in between) and at varying intensity (e.g., shallow versus steep dives). Similarly, lead and lag pursuit can be performed in a number of ways. It is possible for a given action input to accommodate any of the four concepts. As such, concept explanations based on action space are insufficient to describe these concepts which human pilots readily understand stressing the need for higher-level, abstract concept representation.

Notably, $\{Lead, Lag\}$ and $\{Climb, Dive\}$ are physically mutually exclusive within each subset (e.g., an aircraft cannot simultaneously climb and dive at the same time).

## G.4 CONCEPT LABELING PROCEDURE

For each logged state-action sample $(X_i, Y_i)$, we derive concept labels $c_i^{(\ell)}$ directly from the simulator state using simple geometric rules that reflect the operational definitions of lead/lag pursuit and climb/dive in formation training.

**Lead vs. Lag** Lead and lag pursuit are labeled using ground-plane geometry between the lead and chase aircraft. From the simulator state we extract the ground-plane positions $(p_L, p_C)$ and yaw angles $(\psi_L, \psi_C)$, form the corresponding nose-direction rays in the horizontal plane, and solve for their intersection. If the two rays intersect ahead of both aircraft, we label the state as *lead pursuit* (0); otherwise we label it as *lag pursuit* (1). This matches the intuitive notion that the chase aircraft is in lead pursuit when its nose is pointed ahead of the lead's flight path, and in lag pursuit otherwise (Fig. 3).

**Climb vs. Dive** Climb and dive are labeled using the sign of the vertical velocity of the chase aircraft. Given consecutive altitudes $z_{\text{prev}}, z_{\text{now}}$ and time step $\Delta t$, we estimate $v_z = (z_{\text{now}} - z_{\text{prev}})/\Delta t$. States with $v_z \geq 0$ are labeled *climb* (0), and those with $v_z < 0$ are labeled *dive* (1).

All states receive a 0/1 label for each concept. These labels are deterministic functions of $X$ and therefore do not enter the causal target estimator SCM. They are used only to index concept reference sets when estimating interventional effects.

