# OpenReview forum: "Causal Explanations for Human Understanding in Deep Neural Policies"
_ICLR.cc/2026/Conference — Submitted to ICLR 2026_

### Official Review · Reviewer_ba8L · 2025-10-29

**Soundness:** 2
**Presentation:** 1
**Contribution:** 2
**Rating:** 2
**Confidence:** 4

**Summary:**

This paper introduces CCW-Net, a method that aims to enforce causality in concept-based explanations for deep neural policies. To do so, it aligns each concept’s influence on actions with causal effect estimates from data. The method is tested in a realistic aircraft formation task, where it is shown that CCW-Net improves concept-to-action causal alignment while maintaining accuracy.

**Strengths:**

- enforcing causally-aligned explanations into RL is an interesting research direction

- this research direction seems novel

**Weaknesses:**

Major:

- **W1** - The evaluation is minimal:
    - Although the paper is presented as methodological and makes general claims, the evaluation is limited to a single application (aircraft formation). Furthermore, from the paper, it was not clear to me how causality is key for this application.

    - No comparisons against alternative approaches are presented. It seems to me that the proposed method is only compared to an ablation of itself by removing the alignment loss.

    - Broad benefits of the proposed methodology are not studied, nor are metrics specific to causality.


- **W2** - Assuming no unmeasured confounding seems a very strong assumption, which could not apply to practical real-world settings. It seems to me the validity of this assumption is not discussed in the paper, neither theoretically nor empirically.

- **W3** - Notation is very confusing and inconsistent, which makes it hard to evaluate the soundness of the approach. Find below a non-exhaustive list:

    - It is not clear what uppercase and lowercase symbols indicate, e.g., “we extract the latent vector Z”, but then “vector concepts $s_j$”.

    - undefined symbols: e.g., $\tilde{C}$ and $p$ in l.272 are not defined.

    - In the main text, I could not find a formal definition for two of the three employed losses, i.e., $L_{cls}$ and $L_{imit}$.

    - The same symbol $\theta$ is used for both the encoder $h$ and the head $g$.

    - In l. 306, $A$ is used as the dimensionality of a vector space, while it was previously defined as the variable denoting binary concept activation.

    - Concept labels were indicated with $c^{ell}$, but later, in l. 376, “Concept labels $A = (A_L, A_C)$”.

    - The alignment loss is sometimes referred to as ‘causal loss‘, also in Algorithm 1.

     - The chosen naming for variables is unusual:  $C$ is used to represent the raw input $X$, $A$ to represent concept activation, and $M$ for the set of concepts’ representations, for which the notation goes back to $c_j$.

Effort should go in making the notation clear and consistent. To start, I would recommend using uppercase symbols to indicate random variables, lower case (with indices) for realizations and bold to distinguish scalars from vectors.



Minor:

- typo: “with concept labels a tuple” (l.205).
- ll.260-264 repeat the lines above.

**Questions:**

- **Q1** (Related to W1) - What makes the investigated application (aircraft formation) best-suited to test causal explanations?

- **Q2** - I am not sure I understand the part where the authors claim to represent concepts as vectors. Concepts need to be aligned with human semantics; how are these interpretable? Where are these vectors computed?


While I remain open to a constructive discussion, I believe the paper requires substantial improvement, especially in clarity and in the evaluation section. At this stage, I'm not inclined to raise my score, as the gap between the current submission and a version that would meet the bar for acceptance still feels too wide.

---

> ### Author Response · Authors · 2025-11-24
>
> Thank you for your time and candid feedback. It helped us substantially improve the paper. We address each concern in turn:
>
> **Single task:** We agree that multi-task evaluation is often valuable. In our case the aircraft formation task is *especially discriminative* for the problems CCW-Net addresses: making a black-box policy interpretable in terms of human concepts and validating those explanations via necessary-and-sufficient causal attributions. This task is (i) high-dimensional and underactuated, (ii) strongly context-dependent, and (iii) grounded in real operational ontology. These properties break prior art and directly motivated CCW-Net.
>
> Scalar wrapper methods [1,2] work in domains where concepts map cleanly to actions. Here, they fail: scalar concepts cannot reproduce the aircraft policy Extending to vector concepts breaks its causal semantics because influence leaks across other concepts’ degrees of freedom.
>
> Three properties make this task a strong testbed:
>
> - **Real expert concepts:** Long-standing operational ontologies providing external validity absent in synthetic domains.
> - **Minimal, complete ontology.** Despite dynamic complexity, four well-defined concepts fully support the task, avoiding confounds from large/noisy ontologies and ensuring completeness validated by decades of aviation practice.
> - **Ambiguous action-concept mapping:** Identical controls map to different any concept depending on geometry and energy state, stressing causal mediation in a way simpler tasks cannot.
>
> In short, this task is uniquely discriminative for the challenges CCW-Net is designed for: grounding explanations in human concepts and aligning them causally to the policy’s actions. To our knowledge, only interventional vector concepts recover the correct causal pathway in the wrapper setting.
>
> We are exploring additional domains, but we believe this one provides a particularly stringent and meaningful evaluation.
>
> ***
>
> **Causality key for task:** In safety and task-critical settings like aviation, correlational explanations as not enough. If a system says “I turned because I was in lag pursuit,” we need evidence that changing that concept would have changed the action. CCW-Net’s interventional effects provide exactly that standard of causal justification.
>
> ***
>
> **Comparisons to alternatives:** We agree that baselines matter, however, existing concept-wrapper methods are not insufficient comparators for this task. CW-Net [2] with scalar concepts cannot reproduce the aircraft policy, extending it to $d>1$ breaks it causal claim, and PW-Net [1] is correlational.
>
> With the intent to serve as a wrapper on a pre-trained policy (strength identified by jZQw) we see the meaningful baseline as the wrapper with and without causal alignment. We're performing further ablations per the other reviewers.
>
> ***
>
> **Broad benefits:** CCW-Net can be viewed as a form of causally-aligned representation learning (updated discussion in paper). It shapes the concept encoder’s latent geometry using interventional effect estimates rather than predictive correlations. This encourages the directions in concept space to become locally necessary and sufficient to actions. RhgA notes benefits to validation and governance. Interpretable and trustworthy models are primary motivators.
>
> ***
>
> **Causal metrics:** CCW-Net evaluates causal effects using standard metrics from interventional multiple-mediator analysis, $IE_j$, $TE$, $IE_\mu$, which follow from [3,4]. To assess the identification conditions required by these works, we report standard overlap diagnostics (coverage, ESS, fallback).
>
> ***
>
> **No unmeasured confounding:** Thank you for raising this point. In response, we made this assumption explicit and added Proposition 1 (App. C.2), which we developed to prove that mediator-outcome ignorability holds by construction in CCW-Net’s induced SCM. The assumption is not an environmental one but a property guaranteed within the induced SCM.
>
> ***
>
> **Notation and clarity:** Thank you for highlighting this. It was extremely helpful. We now fully standardize the notation and rewrote Sec. 5 which is now organized cleanly around the two SCMs, Fig. 1 simplified, Fig. 2 added, and Apps. A/C/E provide complete expanded information. These changes address the inconsistencies you identified and improve readability.
>
> ***
>
> **Concept vectors and interpretability:** Thank you for raising this. The vector is not the interpretable object. The concept label is. Vectors encode how a human-defined concept (e.g. climb) should be expressed in context. They are computed by the concept encoder $g_\theta$ from $Z$, and their meaning comes from the supervised label. The vector form captures context-dependent variation for imitation.
>
> Thank you for your time! Your feedback has greatly strengthened this work.
>
> [1] Kenny et al., ICLR 2023
>
> [2] Kenny et al., arXiv:2411.18714, 2024
>
> [3] Vansteelandt & Daniel, Biometrics 2012
>
> [4] Loh et al., Psych. Methods 2022

---

### Official Review · Reviewer_jZQw · 2025-10-30

**Soundness:** 2
**Presentation:** 2
**Contribution:** 2
**Rating:** 4
**Confidence:** 4

**Summary:**

This paper introduces Causal Concept-Wrapper Network (CCW-Net), a method for training neural policies that provide human-interpretable explanations grounded in causal relationships. The approach wraps a pre-trained black-box policy with a concept module and policy head, using supervised learning from expert trajectories labeled with human-defined concepts. The method uses mediation analysis techniques to align gradient-based concept attributions with interventional indirect effects estimated from observational data. The authors evaluate CCW-Net on an aircraft formation control task using three different architectures: capsule networks, vector concept bottleneck models, and cross-attention mechanisms. Results demonstrate that the causal alignment loss improves the correspondence between concept sensitivities and causal effects while maintaining task performance and concept classification accuracy.

**Strengths:**

* Using a black-box policy is a practical advantage: you can apply this to existing trained policies without starting over. This is important since retraining can be expensive or impossible in many real systems.

* The experiments are well-organized with clear hypotheses (H1-H4) that are tested one by one. This makes it easy to see what the authors claim and whether the data supports it.

* The method works across three different neural architectures (capsule networks, concept bottleneck models, and cross-attention). This flexibility matters for real-world deployment.

* The aircraft formation task is a good test case because pilots actually use these concepts (lead/lag pursuit, climb/dive) in training and operations. This grounds the evaluation in real interpretability needs rather than made-up concepts.

**Weaknesses:**

My main concern is that the causal framework isn't clearly defined. The paper introduces a structural causal model but it's hard to tell exactly what the variables are and how they relate to each other. The notation switches between (X, Z, M, Y) and (C, A, M, Y) without clear explanation, which makes it difficult to evaluate if the causal claims are valid. The paper mentions standard assumptions for causal identification but doesn't explain how to check if these hold in practice or what happens when they don't.

Minor points:
* Some relevant related work is missing:
  - Work on using causal models for understanding language models [1,2]
  - Work on causal explanations of RL policies [3,4]
  - Explanations of sequential decision making [5,6]

References:

[1] Geiger, Atticus, Chris Potts, and Thomas Icard. "Causal abstraction for faithful model interpretation." arXiv preprint arXiv:2301.04709 (2023)

[2] Atticus Geiger, Hanson Lu, Thomas Icard, and Christopher Potts. Causal abstractions of neural networks. NeurIPS, 2021

[3] Madumal, Prashan, et al. "Explainable reinforcement learning through a causal lens." 2020.

[4] Kekic et al. "Learning Nonlinear Causal Reductions to Explain Reinforcement Learning Policies." (2025).

[5] Bica et al. Learning "what-if" explanations for sequential decision-making. ICLR, 2021.

[6] Nashed, Samer B., et al. "Causal Explanations for Sequential Decision Making." Journal of Artificial Intelligence Research 83 (2025).

**Questions:**

* Definition of the SCM: in L214 it seems like the set of variables (or space of the variables) is given by $\mathcal{X}, \mathcal{Z}, M, \hat{\mathcal{Y}}$, but the samples of the variables are $C, A, M, Y$ and the causal relations are given in terms of $C, A, M, Y$. What are the variables in the SCM? What are the assumed causal relationships?
* What are the assumptions made on the SCM? How did you check that these assumptions are valid in the RL case? In cases where those assumptions do not hold or only hold approximately what are the consequences for the method?
* Assumptions in L144: for all 3 assumptions, given a concrete policy/dataset, how would you check if those assumptions hold? Is there a way to measure this in order to determine if the method is appropriate for a given policy?
* L184 "Furthermore, following Loh et al. (2022), we relax assumptions on the causal graph by introducing an interaction term": which assumptions and why can they be relaxed?
* What is the conceptual difference between $A$ ("which correspond to a binary concept activation, representing if a concept is active or not" L216) and the mediators $M$ ("to represent vectors cj corresponding to each concept" L218)?
* How is this causal formulation different from a purely correlational model? Wouldn't a non-causal model that tries to predict a concept given the current state and action lead to the same results? If not, can you describe a situation in which such two models (mediation analysis model and correlational) would not yield the same concepts?

I'm happy to increase my score if we can resolve these points of confusion during rebuttal.

---

> ### Author Response · Authors · 2025-11-24
>
> Thank you for the time you put into your thorough and thoughtful feedback! We address each point below:
>
> We agree that the ability to wrap an existing high-performing policy without retraining is a practical strength, and that the aircraft formation task is an appropriate, high-fidelity setting because its concepts (lead/lag pursuit, climb/dive) come directly from real operational practice.
>
> ***
>
>   **Framework clarity:** Thank you for noting the confusion. We revised Sec. 5, updated Fig. 1, added Fig. 2, clarified Alg. 1, and reorganized the appendix. The confusion likely stemmed from mixing two notation systems. The revised paper makes our two SCMs explicit using only:
>
> - Wrapper SCM: $(X,Z,M,\hat Y)$
> - Causal target estimator SCM: $(X,Z,M,\tilde Y)$
>
> Both SCMs now appear explicitly in Fig. 2.
>
> ***
>
> **Definition of the SCM:** Figure 2 and Section 5 now explicitly specify the variables and assumed causal relationships. The wrapper SCM contains only the mediated path $X$->$Z$->$M$->$\hat Y$ (no direct $Z$->$\hat Y$ link). The causal target estimator SCM mirrors this structure but uses frozen snapshots so that interventional effect estimation is stable.
>
> ***
>
>   **SCM assumptions:** This question greatly helped us strengthen the paper.
>
> Ignorability $(\tilde Y \perp M | X)$: Usually untestable, but our newly added Proposition 1 (App. C.2) shows this holds *by construction* in the induced SCM whenever:
> - $g^{(ref)}$ and $h^{(ref)}$ are fixed during target estimation,
> - Decoder randomness is independent of the randomness that generates $M$,
> - Matching noise uses an independent RNG stream.
>
> All three are enforced in our implementation (deterministic evaluation mode, frozen snapshots, independent seeds). Thus, *ignorability is guaranteed for the induced SCM*.
>
> Positivity (support of P(M|X)): We provide overlap diagnostics (coverage, fallback rate, accepts/sample, ESS (App. C.5), allowing practitioners to check support empirically.
>
> Consistency: Holds whenever $h^{(ref)}$ is deterministic for a fixed snapshot. Users verify this by evaluating the snapshot in deterministic mode and fixing RNG streams.
>
> We emphasize that CCW-Net's causal claims apply *only to the induced SCM*, not the environment dynamics (saved for future work).
>
> ***
>
> **Relaxed assumptions per Loh et al. (2022):** Interventional indirect effects (IIEs) do *not* require a causal ordering among mediators or mediator independence. All mediator-mediator dependence is absorbed into a residual interaction term $IE_\mu$. This is important because softmax and capsule squash operations naturally induce dependencies between concept vectors.
>
> ***
>
>   **Difference between $A$ and $M$:** Thank you for raising this question. This was ambiguously presented before. We now make the distinction explicit:
> - Binary concept labels $c^{(\ell)}_j$: expert supervision only, used for concept classification and forming reference sets
> - Mediator vectors $M_j$: learned continuous vector representations produced by the concept encoder $g_\theta$, which participate in causal effect estimation and policy decisions.
>
> ***
>
> **Causal formulation vs correlational model:** A correlational model (e.g. predicting concepts from $(X,Y)$ or using gradients/attention) assigns importance to any feature correlated with the action, even if that feature is *not* causally responsible. CCW-Net instead uses *interventional* effects: for each concept $j$, we estimate $IE_j$ by replacing only $M_j$ with a counterfactual mediator while holding all other mediators fixed. The two approaches diverge whenever concepts are correlated but do not lie on the causal path to the action.
>
> **Example from our task:**
> “Climb” often co-occurs with “lag pursuit,” so a correlational model assigns importance to both.
> CCW-Net’s mediator swap shows:
>
> $IE_{lag}$ is large,
>
> $IE_{climb}$ ~=0,
>
> revealing "climb" is predictive but not causally responsible once the other concepts are held fixed
>
> ***
>
> **Further relevant work:** Thank you for these! We’ve incorporated them into our expanded related work section in App. B. These works analyze or validate existing models via counteraction reasoning, SCM abstractions, or causal reductions. We believe our method contributes to the field as it uses interventional mediation effects as training targets to shape the learned concept space itself (aligning concept-to-action sensitivities with causal effects estimated from logged trajectories). CCW-Net complements prior causal-analysis methods by introducing a causally guided representation learning objective rather than a post-hoc causal explanation mechanism.
>
> ***
>
> Thank you for your time and hope this discussion and our manuscript edits clarify your questions.
>
> ***
>
> [1] Stijn Vansteelandt and Tyler J VanderWeele. Natural direct and indirect effects on the exposed: Effect decomposition under weaker assumptions. Biometrics, 68(4):1019–1027, 2012.

---

### Official Review · Reviewer_GNNi · 2025-10-30

**Soundness:** 3
**Presentation:** 3
**Contribution:** 3
**Rating:** 6
**Confidence:** 3

**Summary:**

This paper introduces the CCW-Net, which explains the actions taken by a pre-trained reinforcement algorithm with human interpretable concepts. The paper builds on concept explanations by extending them from scalar valued to vector valued. Furthermore, they add causality based explanations using mediator interactions. This paper has interesting contributions to explainability via concepts, such as moving away from correlation based interactions to causality based interactions. They also demonstrate that, on the aircraft formation task, the casual alignment loss improves the concept-to-action causal alignment across three architectures while not harming the concept accuracy.

**Strengths:**

1.	This paper lays the ground for producing casual explanations in reinforcement learning algorithms using human interpretable concepts.
2.	Demonstrating the increased casual alignment without degradation in performance
3.	Interesting approach on aligning concept representations with casual effect.
4.	The methodology is head-agnostic as long as everything remains differentiable (which is usually the case, and it is an relatively easy constraint to satisfy)
5.	Clearly written paper, easy to follow.

Overall, the paper is well written an easy to follow and could be a nice contribution to the field. Therefore, I am leaning more to acceptance and the score could be increased if further ablation studies are done, limitations are clearly stated, anddifferent tasks could be added.

**Weaknesses:**

1.	Limitations are not stated in the paper as well as the computational requirements for training vector valued concept explanations.
2.	Lack of ablation studies on the loss lambda terms, and the fraction of replacement of reference sets.
3.	Tested on only one task, thus would be nice to see the application of this methodology on a different task.

**Questions:**

For the further ablation studies, it’d be interesting to see the affect the lambda parameter on the alignment. For example, how would the alignment score change with lambda = 0.01, and lambda = 1.0 is used? Are there any drawbacks to using higher lambda? Also I am curious what would the effect of replacing a higher fraction of the Reference sets (p) would be on the alignment scores.

Furthermore, it would be nice to see a different task so that the claims can be further supported and validated.

Further clarifications questions I have:
1.	What is the reason behind excluding the Direct Effects on the Total Effects?
2.	What is the computational complexity of adding more concepts to the pipeline in terms of computational resources and training time?

Remarks:
- Typo in the last line 13 of the algorithm section, should be L_align instead of L_casual
- opening quotes in last line of discussion ``

---

> ### Author Response · Authors · 2025-11-24
>
> Thank you for your time and thoughtful feedback! We address each point below:
>
> **Limitations and computational requirements:**
>
> We added a limitations section. In brief: CCW-Net’s causal claims apply only within the induced SCM (concept->action), not the environment’s dynamics (left for future work). Identification depends on overlap in concept-conditioned reference sets. Poor support increases bias and variance. Explanations are restricted to the chosen concept ontology. If task relevant factors are not included, their influence may be absorbed into degrees of freedom within $M$, not be named as a concept, and not appear in explanations.
>
> Computationally, each batch requires one forward pass through the train-time model, one forward pass through the frozen snapshot as a counterfactual reference, and approximately $J S$ additional frozen-head evaluations for counterfactual mediator hybrids (with $J$ concepts and $S$ Monte Carlo samples), depending on which hybrid combinations are meaningful.
>
> ***
>
> **Ablation studies:** We agree these ablations are important. Experiments varying $\lambda_{align}$ (0.01, 0.1, 1.0) and the reference-set refresh ratio (0.0, 0.2, 0.5, 1.0) are underway. Results will be shared when complete.
>
> ***
>
> **Single task:** We agree that multi-task evaluations are often valuable. In our case, however, the chosen aircraft formation task is uniquely well suited for evaluating CCW-Net and exposes failure modes that simpler tasks cannot. This task is (i) high-dimensional and underactuated, (ii) strongly context-dependent, and (iii) governed by a real-world operational concept ontology used by pilots in practice. These properties motivated CCW-Net’s design.
>
> Prior concept-wrapper methods such as PW-Net and CW-Net [1,2] succeed on simpler domains (e.g. Atari-like control), but fail on this task. Scalar concepts remove too much information to reproduce the expert aircraft policy, and increasing concept dimension to $d>1$ breaks the faithfully causal claim in [2]. These gaps directly motivated CCW-Net’s vector-valued causal mediation approach.
>
> Central to this domain is that all control inputs can express any of the four concepts depending on geometry and relative energy state. This makes correlational concept-action mappings fundamentally ambiguous and creates a particularly stressing setting where, to the best of our knowledge, only interventional, vector-valued concept representations can recover the correct causal pathway. This is exactly the type of regime CCW-Net is designed to address.
>
> The aircraft formation domain has meaningful advantages for causal explainability research:
>
> - Real expert concepts. The selected concepts are not contrived for research but long-standing operational abstractions used in training and operations. Evaluating CCW-Net on *real-world* professional concepts strengthens validity not enjoyed by other tasks (real-world grounding mentioned as an advantage by reviewer jZQw).
> - Ambiguous action-concept mapping. The same control inputs can express any of the four concepts depending on geometry and energy state making correlational explanations insufficient and providing an ideal testbed for causal mediation.
> - Minimal concept set. Despite the task’s dynamic complexity, only four human-interpretable concepts fully support the task. This produces a clean, controlled evaluation of concept-level causal alignment without confounds from either large, noisy, or ill-defined concept vocabularies and reduces the risk of unknown missing concepts. Lead/lag/climb/dive are validated as a sufficient concept set through decades of aviation formation flight.
>
> In short, the aircraft formation task stresses exactly the challenges CCW-Net is designed to address: vector concepts, causal mediation, and disentangling concept meaning in a context-dependent control policy. From a research perspective, the aircraft formation domain offers a uniquely demanding setting to evaluate whether causal mediation and vector-valued concepts are doing meaningful work. We are actively exploring additional domains, but we believe the chosen task provides a particularly stringent and meaningful evaluation of CCW-Net.
>
> ***
>
> **Excluding direct effects:** Thank you for highlighting this question as it lead to improving the clarity of Sec. 5.
>
> The wrapper architecture removes any direct pathway from $X$ or $Z$ to $\hat Y$, enforcing
>
>  $X$->$Z$->$M$->$\hat Y$,
>
> Thus the direct effect is zero by construction, and
>
>   $TE=\sum_j IE_j+IE_\mu$,
>
> is the only valid decomposition. This ensures all explanations flow strictly through concepts.
>
> ***
>
> We thank you for your time and feedback and how it has improved the work!
>
> ***
>
> [1] Kenny et al., ICLR 2023
>
> [2] Kenny et al., arXiv:2411.18714, 2024

---

> > ### Comment · Reviewer_GNNi · 2025-11-28
> >
> > I thanks the authors for their response. I will retain my original assessment (marginally above acceptance threshold).

---

### Official Review · Reviewer_RhgA · 2025-11-01

**Soundness:** 3
**Presentation:** 3
**Contribution:** 3
**Rating:** 4
**Confidence:** 4

**Summary:**

The paper wraps a frozen, high-performing policy with a concept module and lightweight head so that the model’s concept→action sensitivities (local Jacobians) are explicitly aligned to interventional, data-estimated causal effects (via mediator-swap–style estimators). The wrapper is head-agnostic (vector CBM, capsules, cross-attention shown). On an aviation control task, the method improves causal alignment of explanations while preserving imitation error and concept accuracy.

**Strengths:**

* **Originality.** Trains attributions to match interventional effects rather than reporting correlational saliency; vector concepts enrich semantics; wrapper applies across heads.
* **Quality.** Clear objectives, reasonable identifiability story, overlap diagnostics, and consistent gains in causal alignment without degrading task performance.
* **Clarity.** Clean separation of (a) interventional effect estimation, (b) local sensitivity computation, and (c) alignment.
* **Significance.** In safety-critical control, auditable concept-level attributions checked against interventional baselines are valuable for validation and governance.

**Weaknesses:**

1. **Strong supervision requirement (trajectories + expert concepts).**
   The approach requires expert-defined concept ontologies and per-trajectory concept labels. Performance and portability depend on the quality, granularity, and coverage of these concepts; annotation cost and domain expertise create a practical bottleneck.

2. **Sensitivity to data scale & overfitting (imitation learning).**
   The wrapper is trained by supervised imitation. With fewer labeled trajectories and concepts, the head can overfit, yielding optimistic in-sample alignment but poor generalization. Results are reported at a single (large) data scale; learning curves vs data size and train–test gaps are missing.

3. **Assumption sensitivity.**
   The method assumes that the declared concepts capture all important pathways from inputs to actions. If any meaningful pathway bypasses these concepts, the interventional effect estimates used for alignment can be systematically wrong. The paper does not include stress tests—such as removing a concept, merging concepts, or injecting synthetic confounders—to evaluate how sensitive results are to such violations.

**Questions:**

1. **Labels & efficiency.** Briefly describe the labeling pipeline and show learning curves vs labeled fraction of trajectories (e.g., 10/25/50/100%).
2. **Small-data generalization.** Report train vs test imitation error and causal alignment at reduced data sizes.
3. **Concept robustness.** Run a compact stress test: remove/merge one concept, add label noise (≈10–20%), and a placebo concept; summarize impact on alignment.

---

> ### Author Response · Authors · 2025-11-24
>
> We appreciate your time and constructive feedback in reviewing our work! We address each point below:
>
> **Strong supervision requirement (trajectories + expert concepts):**
>
> We agree CCW-Net requires an expert-defined concepts and labels, but we view this as aligned with its goal: explanations that speak the same language as domain practitioners. Expert-defined ontologies are not merely a source of supervision; they’re a practical necessity for interpretability and adoption. Many high-stakes fields (e.g. aviation, healthcare, construction engineering, law) already rely on shared expert ontologies to communicate, audit decisions, and ensure semantic consistency across teams and systems. Using domain-native concepts enables the model to explain itself in the vocabulary practitioners already use, which supports transparency, trust, and integration into existing operations.
>
> This was one major reason we chose the selected case study. The task and the selected concepts human pilots use to communicate, train, and debrief are well established in the real world. We sought out to ask whether it was possible to produce non-correlational interpretability using concepts from an established professional community in such a dynamic scenario.
>
> We view trajectories as likely inexpensive to generate once the expert policy is trained. Reviewer jZQw pointed out the practical advantage: “…you can apply this to existing trained policies without starting over. This is important since retraining can be expensive or impossible in many real systems.”
>
> We do agree that annotation cost potentially creates a practical bottleneck, but auto-labeling opportunities may exist such as the one we leveraged in this work (App. G.4 in the latest version). Even when not, we not only believe that domain-grounded human interpretability is not only worth the cost of annotation, but we’re also excited to advance interpretability options available to the community.
>
> Thus, while supervision is required, it provides crucial alignment between model reasoning and practitioner reasoning. We view this as an asset rather than a bottleneck in real-world deployment.
>
> ***
>
> **Question: Labels & efficiency:**
> We added a clear description of the labeling pipeline (App. G.4). A domain expert provided concept definitions, enabling reliable automatic labeling.
>
> Experiments measuring performance vs labeling fraction (10/25/50/100%) are underway, and we will share results as soon as analysis is complete. We are excited to see these results.
>
> ***
>
> **Sensitivity to data scale & overfitting (imitation learning) + Small-data generalization:**
> We agree that this is important. Additional experiments are underway to report train vs test imitation error and causal alignment at reduced data sizes (i.e. 5k and 50k samples). We will share that analysis (including learning curves vs data size and train-test gaps) as soon as they are complete.
>
> ***
>
> **Assumption sensitivity + Concept robustness:**
> We agree that testing robustness to ontology misspecification is valuable. Additional experiments are underway to report the requested stress tests: (i) removing a concept, (ii) merging concepts, (iii) adding label nose at 10% and 20%, and (iv) adding a placebo concept. We will share that analysis as soon as it is complete.
>
> We also now address in Sec. 6 that if the concept ontology omits a task-relevant causal pathway, the causal effects estimated by CCW-Net are strictly with respect to the induced SCM rather than the environment. In such cases CCW-Net remains well defined, but explanations are necessarily limited to the provided ontology. We anticipate the above results to shine light on this topic.
>
> ***
>
> Thank you for these thoughtful suggestions. Particularly, the requested stress tests hold potential to provide valuable insight into the method and we appreciate the opportunity to strengthen the work!

---

### Meta-Review · Area_Chair_KXcC · 2025-12-11

**Summary:**

The reviewers appreciated the contribution of this work, and especially how it can provide causal concept explanations for trained policies without the need to retrain. However, all reviewers had questions about the limitations and assumptions of the work, as well as how it was demonstrated on a single application with no baseline comparisons. I found reviewer's RhgA questions particularly helpful to improve the work.

**Reviewer Concerns:**

The authors have responded to each reviewer point by point and promised ablations and further experiments in some of their answers. While some of their answers were convincing, I could not find evidence of the new analyses in the revision or in comments. Therefore, the authors' response is somewhat limited.

**Reviewer Scores:**

This work received mixed reviews, with one more favourable score and a rejection score and both reviewers mentioning they were unlikely to modify their evaluation.
Given the lack of further ablations and experiments, especially to reviewer's RhgA points, I do not believe the reviewers with less positive scores would have modified their recommendation and I think the work would benefit from these additional experiments before publication.

---

### Decision · Program_Chairs · 2026-01-26

Reject